# Differentially Private Relational Learning with Entity-level Privacy Guarantees

**Yinan Huang**[*]
Georgia Institute of Technology
yhuang903@gatech.edu

**Haoteng Yin**[*]
Purdue University
yinht@acm.org

**Eli Chien**
Georgia Institute of Technology
ichien6@gatech.edu

**Rongzhe Wei**
Georgia Institute of Technology
rongzhe.wei@gatech.edu

**Pan Li**
Georgia Institute of Technology
panli@gatech.edu

## Abstract

Learning with relational and network-structured data is increasingly vital in sensitive domains where protecting the privacy of individual entities is paramount. Differential Privacy (DP) offers a principled approach for quantifying privacy risks, with DP-SGD emerging as a standard mechanism for private model training. However, directly applying DP-SGD to relational learning is challenging due to two key factors: (i) entities often participate in multiple relations, resulting in high and difficult-to-control sensitivity; and (ii) relational learning typically involves multistage, potentially coupled (interdependent) sampling procedures that make standard privacy amplification analyses inapplicable. This work presents a principled framework for relational learning with formal entity-level DP guarantees. We provide a rigorous sensitivity analysis and introduce an adaptive gradient clipping scheme that modulates clipping thresholds based on entity occurrence frequency. We also extend the privacy amplification results to a tractable subclass of coupled sampling, where the dependence arises only through sample sizes. These contributions lead to a tailored DP-SGD variant for relational data with provable privacy guarantees. Experiments on fine-tuning text encoders over text-attributed network-structured relational data demonstrate the strong utility-privacy trade-offs of our approach. Our code is available at https://github.com/Graph-COM/Node_DP.

## 1 Introduction

Complex real-world relationships and interactions among individuals are commonly modeled as graphs, where nodes represent these individuals (or more broadly, entities) and edges depict their connections. These graph structures are invaluable for analyzing intricate systems and networks [1, 2, 3, 4, 5, 6]. Machine learning methods that leverage these structures have demonstrated significant potential in tasks reliant on such relational information, such as recommendation systems, financial network analysis, and some healthcare applications [7, 8, 9, 10, 11, 12, 13]. Particularly, relational learning integrates relational information into the models, enabling them to capture and infer complex dependencies beyond isolated data points [14, 15, 16]. One promising application is fine-tuning foundation models—originally trained on text or images—using relational datasets to improve their predictive performance and adaptability to relational contexts [17, 18, 19, 20].

Privacy concerns emerge when relational data to be incorporated into models contains sensitive information, particularly in healthcare and finance. For example, in a patient-medical record network

---

[*]Equal contributions, listed in alphabetical order.

39th Conference on Neural Information Processing Systems (NeurIPS 2025).

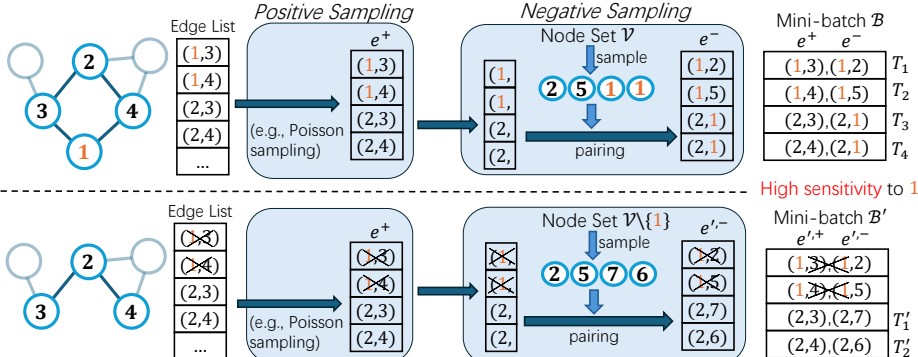

Figure 1: The mini-batch sampling process in relational learning is a two-stage sampling that first samples edges from full edge list (positive sampling) and then construct negative edges based on the sampled positive edges (negative sampling). A node can occur in multiple positive and negative edges, leading to a high sensitivity. Note than the entire batch changes due to the removal of node 1.

data, patient diagnoses (entity attributes) and medical treatments (relations) represent confidential details. Machine learning models are susceptible to privacy vulnerabilities, potentially leading to unintended exposure of sensitive training data [21, 22].

Differential Privacy (DP) provides a principled framework to quantify and mitigate privacy risks for individual data [23]. A prominent approach to train deep learning models under the DP framework is DP-SGD [24, 25, 26, 27, 28, 29, 30]. DP-SGD was initially developed for unstructured data with independent samples, where modifying a single data point impacts only one gradient term. Privacy preservation is thus achieved by controlling gradient sensitivity via per-sample gradient clipping and adding calibrated Gaussian noise to mask individual contributions and prevent privacy leakage.

Moreover, DP-SGD benefits from a privacy analysis technique known as privacy amplification by subsampling [26, 31, 32, 33]. Specifically, applying a differentially private algorithm to a randomly subsampled dataset rather than the entire dataset strengthens privacy guarantees. This mechanism aligns with mini-batch sampling, further reinforcing the privacy protection capability of DP-SGD.

However, deploying DP-SGD in relational learning to protect entity-level privacy presents unique challenges due to the interconnected nature of entities and the relationship-oriented training paradigm (Figure 1). These challenges include: a) **high sensitivity**: entities often participate in multiple relations, thereby appearing in *multiple loss terms*, significantly increasing sensitivity to entity removal and the risk of sensitive information leakage; and b) **coupled sampling**: mini-batches in relational learning typically involve multiple interdependent sampling steps, such as sampling positive relations first and subsequently generating negative relations conditioned on these positives [34, 35]. This coupled sampling approach diverges from the standard single-step independent sampling assumed by traditional DP-SGD analyses, complicating the direct application of established privacy guarantees.

This work addresses entity-level private relational learning with the following contributions:

- We provide a gradient sensitivity analysis for the general contrastive loss used in relational learning, considering scenarios where nodes participate in multiple loss terms via multiple positive or negative relations. To mitigate this high sensitivity, we propose an adaptive gradient clipping method that adjusts gradient clipping thresholds based on node occurrences within mini-batches.

- We formalize privacy amplification under coupled sampling. While general coupled sampling poses significant analytical challenges, we identify a tractable subclass where coupling arises solely from sample sizes across multiple stages. We derive corresponding amplification bounds for this setting and show that mini-batch construction in relational learning can be aligned with this structure through carefully designed negative sampling methods.

- Integrating the above ideas, we propose a DP-SGD variant tailored for relational learning with entity-level privacy guarantees. This variant incorporates precise sensitivity control and rigorous privacy bounds. We demonstrate its effectiveness by fine-tuning pre-trained text encoders on text-attributed network-structured datasets, highlighting promising utility-privacy trade-offs.

## 2 Related Works

**Differentially Private Network Analysis and Learning.** Extensive research has focused on developing privacy-preserving algorithms for network analysis and learning under differential privacy (DP) guarantees [36, 37]. [38] studied differentially private community detection for stochastic block models. [39, 40] considered the privatization of the graph Laplacian spectrum. [41] proposed a locally differentially private algorithm for graph neural networks (GNNs). For node-level learning tasks, [42, 43, 44, 45, 46] developed DP methods for training GNNs. [47] employed a teacher-student framework to support differentially private GNN model release. Note that existing DP GNN methods are **not applicable** due to the distinct challenges of relational learning. These methods primarily address node label prediction tasks, with their privacy mechanisms focused on preventing leakage of node features during neighborhood aggregation, where the interaction data is used as the input features to the model. In contrast, the central privacy risk in relational learning stems from the relations themselves being part of the loss function computation, i.e., the loss format, where mini-batches are constructed through complex, coupled sampling of relations, and a single entity can participate in numerous such relations.

**Privacy Amplification by Subsampling.** Privacy amplification refers to an enhanced privacy guarantee that comes from subsampling the raw dataset [48, 49, 50]. Most existing analyses focus on neighboring datasets that differ by a single data point [31, 32, 51, 33, 52]. Recently, it has been extended to group privacy, allowing neighboring datasets to differ by multiple data points [53, 54, 55]. These works commonly assume the dataset is an unstructured collection of independent data points, and the sampling mechanism simply outputs a subset of it. Our work instead extends the sampling mechanisms to coupled sampling, an interdependent sampling process from a composite dataset.

## 3 Preliminary

**Entity/Node-level Differential Privacy.** We use graph notations to denote the relationships between entities. Let $\mathcal{G} = (\mathcal{V}, \mathcal{E}, X)$ be a undirected graph defined on node (entity) set $\mathcal{V}$ and edge (relation) set $\mathcal{E}$. Node attributes $X$ assign each node $v \in \mathcal{V}$ an attribute $x_v$. The notion of differential privacy (DP) or Renyi DP (RDP) is defined based on neighboring unstructured dataset differed by one data point [23, 56]. To extend it to graphs, neighboring graphs are defined correspondingly.

**Definition 3.1.** *Two graphs $\mathcal{G}, \mathcal{G}'$ are called **(node-level) neighboring**, denoted by $\mathcal{G} \sim \mathcal{G}'$, if they are differed in a single node and all its associated edges (by removing or inserting a node).*

**Definition 3.2.** *An algorithm $M$ is called node-level $(\varepsilon, \delta)$-DP, if for all neighboring graphs $\mathcal{G} \sim \mathcal{G}'$, and any output subset $S \subset Range(M)$, it satisfies $\mathbb{P}(M(\mathcal{G}) \in S) \le e^{\varepsilon} \mathbb{P}(M(\mathcal{G}') \in S) + \delta$.*

**Definition 3.3.** *An algorithm $M$ is called node-level $(\alpha, \varepsilon)$-RDP, if for all neighboring graphs $\mathcal{G} \sim \mathcal{G}'$, it satisfies $\frac{1}{\alpha} \log \mathbb{E}_{z \sim Q}(P(z)/Q(z))^{\alpha} \le \varepsilon$, where $P, Q$ are the distributions of $M(\mathcal{G}), M(\mathcal{G}')$.*

**DP-SGD for Private Learning.** DP-SGD is a variant of the standard SGD algorithm, designed to ensure that the training process of a machine learning model adheres to differential privacy [24, 26, 25]. It assumes that the mini-batch gradient is decomposable with respect to each training sample: for a mini-batch $\mathcal{B} = \{x_1, \dots, x_b\}$, the gradient satisfies $\mathbf{g}(\mathcal{B}) = \sum_{x_i \in \mathcal{B}} \mathbf{g}(x_i)$. This assumption enables bounding the sensitivity $\max_{\mathcal{B} \sim \mathcal{B}'} \|\mathbf{g}(\mathcal{B}) - \mathbf{g}(\mathcal{B}')\|_2 \le C$ via per-sample gradient clipping, where each $\mathbf{g}(x_i)$ is replaced by $\mathbf{g}(x_i)/\max(\|\mathbf{g}(x_i)\|, C)$. We refer to this per-sample, independently applied clipping as standard clipping throughout the paper.

**Privacy Amplification by Subsampling.** Let $D$ be a generic dataset, and $S$ be a sampling mechanism that produces a random mini-batch $\mathcal{B} = S(D)$. Traditionally, the dataset is a unstructured collection of samples $D = \{x_1, \dots, x_N\}$, and $S(D)$ a random subset $\mathcal{B} = \{x_{s_1}, \dots, x_{s_b}\} \subseteq D$, e.g., $S$ is sampling without replacement or Poisson sampling. In relational learning, however, $D$ is a graph $\mathcal{G}$ and $S$ involves a complex, multi-stage sampling process over nodes and edges, which we will elaborate in Section 4.2. Let $M \circ S$ denote the mechanism that first applies sampling to the dataset and then runs a randomized algorithm $M$. Given a $(\varepsilon, \delta)$-DP algorithm $M$, the goal of privacy amplification is to estimate the privacy parameters $(\varepsilon', \delta')$ for the composed mechanism $M \circ S$ [52].

# 4   Problem Formulation and Main Results

Relational learning aims to develop node attribute encoders for predicting and inferring relationships between nodes [57, 58, 59]. A key advantage is that once trained, these refined encoders can be deployed for new, unseen entities to predict potential relationships in a zero-shot manner. This capability is particularly valuable in applications such as addressing zero-shot recommendation systems [60, 61, 62, 63], and anomaly detection, where interactions involving the new entities are inherently sparse [64, 65]. Formally, suppose a node attribute encoder $f_\Theta$ maps entity attributes $x_u$ to a feature vector $h_u = f_\Theta(x_u)$. The goal of relational learning is to leverage the edge set $\mathcal{E}$ to learn the parameters $\Theta$. We use a scoring function $\text{score}_{(u,v)} = \text{score}(h_u, h_v)$ that reflects the likelihood of an edge existing between entities $u$ and $v$. To provide supervision signals, we use both positive (observed) edges $e^+ \in \mathcal{E}$ and negative (missing) edges $e^- \notin \mathcal{E}$, typically grouped into edge tuples $T_i = (e_i^+, e_{i,1}^-, \ldots, e_{i,k_{\text{neg}}}^-)$ consisting of one positive edge versus $k_{\text{neg}}$ negative edges [66]. Positive edges are usually sampled from $\mathcal{E}$, while various strategies exist for sampling negative edges [34, 35]. An illustration of the mini-batch sampling process is provided in Figure 1. Once the edge tuples are constructed, training proceeds by minimizing a loss function $\mathcal{L}(\Theta, T_i)$ that encourages higher scores for positive edges and lower scores for negative ones. Common loss functions include the InfoNCE loss, $\mathcal{L}(\Theta, T_i) = -\log\left(\exp\left(\text{score}_{e_i^+}\right) / \sum_{e \in T_i} \exp(\text{score}_e)\right)$ [67], and the Hinge loss, $\mathcal{L}(\Theta, T_i) = \sum_{j=1}^{k} \max(0, \gamma - \text{score}_{e_i^+} + \text{score}_{e_{i,j}^-})$, with margin parameter $\gamma$ [68, 69, 70, 71]. For conceptual simplicity, our discussion focuses on binary relations where edges are either present or absent, though the approach can be extended to multi-relational settings.

Consider a mini-batch of edge tuples $\mathcal{B} = \{T_1, ..., T_b\}$, whose gradient $\mathbf{g}(\mathcal{B})$ has the general form:

$$\mathbf{g}(\mathcal{B}) = \sum_i \mathbf{g}(T_i) = \sum_i \mathbf{g}(e_i^+, e_{i,1}^-, e_{i,2}^-, ..., e_{i,k_{\text{neg}}}^-). \tag{1}$$

Eq. (1) fundamentally departs from the standard DP-SGD setting, as the sensitive data *unit*—a node, in this case—may appear in multiple edge tuples $T_i$, thereby influencing several gradient terms $\mathbf{g}(T_i)$. This setup is also technically distinct from group differential privacy [53], since a node's presence in positive versus negative edges can yield different contributions to the overall sensitivity. A detailed discussion on this point and sensitivity analysis, along with an adaptive gradient clipping method for sensitivity control, will be presented in Section 4.1.

Beyond the sensitivity issue, another key challenge is that $\mathcal{B}$ cannot be interpreted as a direct sample (i.e., a subset) from the dataset $D$. Instead, $\mathcal{B}$ consists of edge tuples composed of both positive and negative edges, each sampled from distinct mechanisms (e.g., from the set of observed edges and from randomly paired nodes, respectively). Moreover, depending on the specific negative sampling strategy employed, these two sampling processes are typically mutually dependent. In Section 4.2, we formalize this sampling procedure as *coupled sampling*, a novel and generally challenging setting for privacy analysis. One of our main contributions is to provide a privacy amplification bound for a subclass of coupled sampling where the coupling arises solely from sample size constraints across multiple stages. Notably, one such coupled sampling strategy can be employed in practice and achieves reasonably good utility in relational learning.

## 4.1   Sensitivity Analysis

Let us fix a node $u^*$ to be removed and analyze the local sensitivity $\|\mathbf{g}(\mathcal{B}) - \mathbf{g}(\mathcal{B}')\|$ where $\mathcal{B}'$ is a neighboring mini-batch of $\mathcal{B}$ by removing a node $u^*$ from $\mathcal{B}$. Formally, we define neighboring mini-batches $\mathcal{B} \sim \mathcal{B}'$ as follows: let $\mathcal{B} = \{T_1, \ldots, T_b\}$ where each edge tuple is of the form $T_i = (e_i^+, e_{i,1}^-, \ldots, e_{i,k_{\text{neg}}}^-)$. If node $u^*$ appears in the positive edge $e_i^+$, then the whole tuple $T_i$ is removed in $\mathcal{B}'$; If, instead, node $u^*$ appears only in a negative edge $e_{i,j}^-$ and not in $e_i^+$, then $\mathcal{B}'$ contains the corroponding tuple $T_i$ but replaces all occurrences of $u^*$ with arbitrary nodes sampled from the graph, as shown in Fig. 1. We adopt this definition of neighboring mini-batches because it appears in our later privacy amplification analysis, corresponding to the dominating pairs that capture the statistical divergence of the subsampled mechanism when applied to neighboring graphs.

For a mini-batch $\mathcal{B}$, we define: $\mathcal{B}_+(u) = \{T_i \in \mathcal{B} : u \in e_i^+\}$ as edge tuples whose positive edges contain node $u$, $\mathcal{B}_-(u) = \{T_i \in \mathcal{B} : u \notin e_i^+, \exists j \in [k_{\text{neg}}], u \in e_{i,j}^-\}$ as edge tuples whose positive edges do not involve node $u$ but negative edges do, and $\mathcal{B}_0(u) = \mathcal{B} \setminus (\mathcal{B}_+(u) \cup \mathcal{B}_-)$. Given the

---

**Algorithm 1** Frequency-based Adaptive Gradient Clipping (FREQ-CLIP)

---

**Input:** Batch $\mathcal{B} = \{T_1, ..., T_b\}$ where $T_i = (e_i^+, e_{i,1}^-, ..., e_{i,k_{\text{neg}}}^-)$, gradient terms $\{\mathbf{g}(T_1), ..., \mathbf{g}(T_b)\}$ and gradient norm clipping threshold $C$.
Find the set of nodes $\mathcal{V}_{T_i}$ occurred in each edge tuple $T_i$ and let $\mathcal{V}_{\mathcal{B}} = \bigcup_i \mathcal{V}_{T_i}$.
**for** $v \in \mathcal{V}_{\mathcal{B}}$ **do**
    $\text{freq}(v) \leftarrow \sum_{T_i \in \mathcal{B}} \mathbb{I}(v \in e_i^+ \text{ or } v \in \bigcup_{j=1}^{k_{\text{neg}}} e_{i,j}^-)$            ▷ Compute $|\mathcal{B}_+(v)| + |\mathcal{B}_-(v)|$
**for** $T_i$ in $\mathcal{B}$ **do**
    $\text{max-freq}(T_i) \leftarrow \max_{v \in \mathcal{V}_{T_i}} \text{freq}(v)$
    $\bar{\mathbf{g}}(T_i) \leftarrow \mathbf{g}(T_i)/(\max\{1, 2 \cdot \text{max-freq}(T_i)\|\mathbf{g}(T_i)\|/C\}$
**Return** $\bar{\mathbf{g}} = \sum_i \bar{\mathbf{g}}(E_i)$                 ▷ Clipped mini-batch gradient with sensitivity $C$

---

removal node $u^*$, we can partition $\mathcal{B}$ into three parts: $\mathcal{B} = \mathcal{B}_0(u^*) \cup \mathcal{B}_+(u^*) \cup \mathcal{B}_-(u^*)$. Similarly, mini-batch $\mathcal{B}'$ can be partitioned into two parts: $\mathcal{B}' = \mathcal{B}_0(u^*) \cup \mathcal{B}'_-(u^*)$, where $\mathcal{B}'_-(u^*)$ shares the same positive edges as $\mathcal{B}_-(u^*)$, but every node $u^*$ in $\mathcal{B}_-(u^*)$ is replaced by some other nodes in $\mathcal{B}'_-(u^*)$. By applying triangle inequality, the local sensitivity w.r.t. removal of $u^*$ becomes

$$\|\mathbf{g}(\mathcal{B}) - \mathbf{g}(\mathcal{B}')\| \leq \sum_{T_i \in \mathcal{B}_+(u^*)} \|\mathbf{g}(T_i)\| + \sum_{T_i \in \mathcal{B}_-(u^*)} \|\mathbf{g}(T_i)\| + \sum_{T'_i \in \mathcal{B}'_-(u^*)} \|\mathbf{g}(T'_i)\|. \quad (2)$$

Equation (2) is technically more difficult to analyze, as positive part and negative part jointly contributes to the sensitivity with different forms of impact.

In standard DP-SGD, a common practice for controlling mini-batch sensitivity is to apply per-sample gradient clipping with a fixed constant i.e., force $\|\mathbf{g}(T_i)\| \leq C$ for any $T_i$. Eq.(2) implies this induces a local sensitivity $(|\mathcal{B}_+(u^*)| + 2|\mathcal{B}_-(u^*)|) \cdot C$ w.r.t. the removal of $u^*$. Although $|\mathcal{B}_+(u^*)|$ could be bounded by maximal node degree, there is no guarantee for $|\mathcal{B}_-(u^*)|$. At the worst case, node $u^*$ could appear in every $T_i$ via negative edges (while not appeared in any positive edges), resulting in a global sensitivity of $2 \cdot |\mathcal{B}| \cdot C$, which is too large to afford.

**Adaptive Clipping.** A key observation is that there is no fundamental obstacle to defining a dynamic clipping threshold, i.e., forcing $\|\mathbf{g}(T_i)\| \leq C(T_i, \mathcal{B})$ where the threshold $C(T_i, \mathcal{B})$ can depend on the edge tuple $T_i$ to be clipped and the whole mini-batch $\mathcal{B}$. Intuitively, when a mini-batch $\mathcal{B}$ is sampled, some nodes may exhibit high sensitivity (appearing frequently), while others may have low sensitivity (appearing infrequently). By allowing the clipping threshold to vary accordingly, we can tailor the gradient clipping to the actual sensitivity profile of the batch, rather than always accounting for the worst-case scenario as in constant clipping. Proposition 4.1 demonstrates that a simple adaptive clipping strategy (implemented by Algorithm 1) can effectively reduce sensitivity.

**Proposition 4.1** (Local sensitivity of adaptive clipping). *For any $\mathcal{B} = \{T_1, \ldots, T_b\}$, define clipped gradient $\bar{\mathbf{g}}(T_i) = \mathbf{g}(T_i)/\max\{1, \|\mathbf{g}(T_i)/C(T_i, \mathcal{B})\|\}$ with $C(T_i, \mathcal{B}) := C/(\sup_{u \in T_i} |\mathcal{B}_+(u)| + |\mathcal{B}_-(u)|)$. Then for any pair of neighboring mini-batches $\mathcal{B}' \sim \mathcal{B}$ that differs with respective to a node $u^*$, it satisfies $\|\bar{\mathbf{g}}(\mathcal{B}) - \bar{\mathbf{g}}(\mathcal{B}')\| \leq (1 + |\mathcal{B}_-(u^*)|) \cdot C$.*

**Constant global sensitivity by restricting $|\mathcal{B}_-(u)|$.** Proposition 4.1 implies that if we can restrict $|\mathcal{B}_-(u^*)|$ by some constant for any $u^*$, then the local sensitivity can also be bounded by a constant for any node $u^*$ for removal, thus bounding the global sensitivity $\sup_{\mathcal{B} \sim \mathcal{B}'} \|\mathbf{g}(\mathcal{B}) - \mathbf{g}(\mathcal{B}')\|$ by the same constant. This can be achieved by choosing any negative edge sampling algorithms that every node appears at most once in the negative edges of the minibatch, yielding $|\mathcal{B}_-(u^*)| \leq 1$.

**Comparison with Standard Clipping.** Compared to standard clipping by $C$, adaptive clipping (Algorithm 1) reduces sensitivity and thus requires less noise for the same privacy budget, but at the cost of "penalizing" the gradients of high-degree nodes due to their potentially large $\mathcal{B}_+(u)$. We argue that this penalty may not significantly harm relational learning in practice. As positive edges are sampled uniformly, high-degree nodes are more frequently included and thus more prone to overfitting. Down-weighting their gradients can help mitigate overfitting and improve generalization to low-degree nodes. Our empirical studies in Section 5.2 shows that adaptive clipping indeed has a better utility-privacy trade-off than standard gradient clipping.

**Remark.** Adaptive clipping in DP-SGD has been extensively studied for non-relational learning [72], which typically adjusts the clipping threshold based on statistics of individual gradient norms, such

---

**Algorithm 2** Negative Sampling Without Replacement (NEG-SAMPLE-WOR)

---

**Input:** A mini-batch of positive edges $E^+ = \{e_1^+, e_2^+, ..., e_b^+\}$, number of negative samples per positive edge $k_{\text{neg}}$, and node set $\mathcal{V}$.
Randomly sample $b \cdot k_{\text{neg}}$ nodes $\{v_{1,1}, ..., v_{1,k_{\text{neg}}}, v_{2,1}, ..., v_{b,k_{\text{neg}}}\}$ without replacement from $\mathcal{V}$.
**for** $v_{i,j}$ in $\{v_{1,1}, ..., v_{1,k_{\text{neg}}}, v_{2,1}, ..., v_{b,k_{\text{neg}}}\}$ **do**
    $w \leftarrow$ a random end in $e_i^+$, and $e_{i,j}^- \leftarrow (w, v_{i,j})$       ▷ pair $w$ with the sampled node $v_{i,j}$
**Return** $\mathcal{B} = \{T_1, T_2, ..., T_b\} = \{(e_i^+, e_{i,1}^-, ..., e_{i,k_{\text{neg}}}^-)\}_{i=1,...,b}$

---

as quantiles [73] or distributional moments [74]. In contrast, our approach for relational learning employs a normalization strategy based on node occurrences within a batch to address the unique challenges of this setting.

## 4.2   Coupled Sampling and Its Privacy Amplification

We study the privacy amplification of noisy gradient $\mathbf{g}(\mathcal{B}) + \mathcal{N}(0, \sigma^2 \mathbf{I})$ with a randomly drawn mini-batch $\mathcal{B}$ in relational learning. The mini-batch is constructed by two steps of sampling: sampling of positive edges $e_i^+$ and sampling of negative edges $e_{i,1}^-, ..., e_{i,k_{\text{neg}}}^-$. We formalize this problem in a general manner, by introducing an abstract, two-step sampling called coupled sampling.

**Definition 4.1** (Coupled Sampling). *Let $D = (D^{(1)}, D^{(2)})$ be a composite dataset consisting of two datasets $D^{(1)}, D^{(2)}$ possibly from different domains. A coupled sampling mechanism $S$ consists of two steps of sampling defined by the following procedure: (1) sample a subset $\mathcal{B}^{(1)}$ from $D^{(1)}$: $\mathcal{B}^{(1)} \sim p(\cdot | D^{(1)})$ with $\mathcal{B}^{(1)} \in 2^{D^{(1)}}$; (2) sample $\mathcal{B}^{(2)}$ conditioned on $D^{(2)}$ and $\mathcal{B}^{(1)}$: $\mathcal{B}^{(2)} \sim q(\cdot | D^{(2)}, \mathcal{B}^{(1)})$; (3) the resulting mini-batch is $\mathcal{B} = S(D) = \{(\mathcal{B}_i^{(1)}, \mathcal{B}_i^{(2)}) : i = 1, ..., |\mathcal{B}^{(1)}|\}$.*

Here coupling refers to the dependency of $\mathcal{B}^{(2)}$ on $\mathcal{B}^{(1)}$. Coupled sampling is a general framework that covers most relational learning sampling settings. For instance, a common practice is to first sample some positive edges and "perturb" one end of these positive edges to form negative edges [68]. In this case, $D^{(1)} = \mathcal{E}$, $D^{(2)} = \mathcal{E}^c$ (non-edges), and: (1) $p(\mathcal{B}^{(1)}|D^{(1)}) = p(e_1^+, e_2^+, ...|\mathcal{E})$ randomly sample a few positive edges from $\mathcal{E}$ (e.g., Poisson sampling); (2) to sample negative edges $\mathcal{B}_i^{(2)} = (e_{i,1}^-, ..., e_{i,k_{\text{neg}}}^-)$, first randomly sample one end of $e_i^+$, denoted by $\tilde{u}_i$, and then sample $k_{\text{neg}}$ negative edges $\mathcal{B}_i^{(2)}$ from $\{(\tilde{u}_i, v) : (\tilde{u}_i, v) \in \mathcal{E}^c\}$, i.e., non-edges localized at $\tilde{u}_i$; (3) the resulting mini-batch is $\mathcal{B} = \{(e_i^+, e_{i,1}^-, ..., e_{i,k_{\text{neg}}}^-) : i = 1, ..., |\mathcal{B}^{(1)}|\}$.

Coupled sampling extends the existing privacy amplification framework by generalizing the sampling process to a multi-step, dependent procedure, where the flexible and potentially intricate dependencies between steps pose new analytical challenges. In the absence of such dependencies, that is, $\mathcal{B}^{(2)} \sim q(\cdot | D^{(2)})$, the contributions of each sampling step can be decomposed, allowing existing privacy amplification results to be adapted: in this decoupled case, the mini-batch $\mathcal{B} = \{(\mathcal{B}_i^{(1)}, \mathcal{B}_i^{(2)})\}_{i=1,2,...}$ can be viewed as a single-step sampling from the Cartesian product dataset $D^{(1)} \times D^{(2)}$. However, when dependencies are introduced, this separation no longer holds.

**Cardinality-dependent Sampling.** While general coupled sampling can be complex, there exists a tractable subclass where the coupling arises solely from cardinality constraints. That is, the sampling of $\mathcal{B}^{(2)}$ depends only on the cardinality of $\mathcal{B}^{(1)}$, and not on its specific contents: $q(\mathcal{B}^{(2)} | D^{(2)}, \mathcal{B}^{(1)}) = q(\mathcal{B}^{(2)} | D^{(2)}, |\mathcal{B}^{(1)}|)$.

In the context of relational learning, cardinality-dependent sampling can be realized by choosing negative sampling methods that are independent of the contents of the sampled positive edges. For instance, we adopt Poisson subsampling as positive sampling, and adopt Algorithm 2 for negative sampling, which first randomly draws some nodes from the node set (independent of specific positive edges), and then constructs contrastive pairs by pairing one end of positive edges with the sampled nodes. Note that this pairing may occasionally produce negative edges that are actually true positives—a known issue in node pairing methods like in-batch sampling [35]. However, this occurs with a negligible probability, especially given the sparsity of real-world graphs.

---

**Algorithm 3** Relational Learning with Entity-level Differential Privacy

---

**Input:** node attribute encoder $f_\Theta$, graph $\mathcal{G} = (\mathcal{V}, \mathcal{E}, X)$, loss function $\mathcal{L}$, maximal node degree $K$, learning rate $\eta_t$, batch size $b$, number of negative samples per positive sample $k_{\text{neg}}$, gradient norm clipping threshold $C$, noise multiplier $\sigma$.

**Initialize** Randomly drop edges from $\mathcal{E}$, so that maximal node degree is capped by $K$ and obtain $\bar{\mathcal{E}}$ [42, Algorithm 1]. Let positive sampling rate $\gamma \leftarrow b/|\bar{\mathcal{E}}|$.        $\triangleright$ Node degree capping

**for** $t = 1$ **to** $T$ **do**
   positive edges $E^+ \leftarrow$ independently choosing each positive relation from $\bar{\mathcal{E}}$ with probability $\gamma$.
   Generate mini-batch edge tuples $\mathcal{B}_t = \{T_1, T_2, ...\} \leftarrow$ NEG-SAMPLE-WOR$(E^+, k_{\text{neg}}, \mathcal{V})$
   $\mathbf{g}_t(T_i) \leftarrow \partial \mathcal{L}(\Theta_t, T_i)/\partial \Theta_t$
   $\bar{\mathbf{g}}_t \leftarrow$ FREQ-CLIP$(\mathcal{B}_t, \{\mathbf{g}_t(T_1), \mathbf{g}_t(T_2), ...\}, C)$      $\triangleright$ Adaptive clipping Algorithm 1
   $\tilde{\mathbf{g}}_t \leftarrow \frac{1}{b} \left[ \bar{\mathbf{g}}_t + \mathcal{N}(0, \sigma^2 C^2 \mathbf{I}) \right]$
   $\Theta_{t+1} \leftarrow \Theta_t - \eta_t \tilde{\mathbf{g}}_t$
**Output** $\Theta_T$

---

Treating the above pairing process as a post-processing operation, the entire coupled sampling can be viewed as a cardinality-dependent sampling with $D^{(1)} = \mathcal{E}, D^{(2)} = \mathcal{V}$: (1) $\mathcal{B}^{(1)} \sim \text{Poisson}_\gamma(\mathcal{E})$ includes each positive edge independently with probability $\gamma$; (2) $\mathcal{B}^{(2)} \sim \text{Sample}_{\text{WOR}}(\mathcal{V}; k_{\text{neg}} \cdot |\mathcal{B}^{(1)}|)$ draws nodes without replacement of size $k_{\text{neg}} \cdot \mathcal{B}^{(1)}$ from the node set $\mathcal{V}$.

The negative sampling Algorithm 2 has advantages in two folds: (1) it decouples negative and positive samplings to cardinality-dependent only; (2) it also restricts $|\mathcal{B}_-(u)| \leq 1$ for any node $u$, which is crucial to control the sensitivity as discussed in Section 4.1. Though there may exist other designs of negative sampling algorithms that also satisfy both properties, we adopt node sampling without replacement due to its implementation simplicity, as well as its regular sample size that allows to efficiently perform the negative edge pairing (e.g., compared to Poisson sampling).

Here, we provide the first privacy amplification bound for cardinality-dependent sampling. For the notion of neighboring mini-batches, we adopt removals/insertions of $K$ data points for $\mathcal{B}^{(1)}$ (denote by $\mathcal{B}^{(1)} \sim_{\Delta,K} \mathcal{B}^{(1),\prime}$) and replacement of one data point for $\mathcal{B}^{(2)}$ (denoted by $\mathcal{B}^{(2)} \sim_r \mathcal{B}^{(2),\prime}$). These neighboring notions align with the Poisson subsampling for $\mathcal{B}^{(1)}$ and the sampling without replacement for $\mathcal{B}^{(2)}$. The parameter $K$ accounts for the fact that multiple associated edges can be removed due to the removal of a node. Also note that neighboring definition $\mathcal{B}^{(1)} \sim_{\Delta,K} \mathcal{B}^{(1),\prime}$, $\mathcal{B}^{(2)} \sim_r \mathcal{B}^{(2),\prime}$ matches the neighboring definition $\mathcal{B} \sim \mathcal{B}'$ in Section 4.1.

**Theorem 4.1.** *Consider a composite dataset $D = (D^{(1)}, D^{(2)})$ of size $|D^{(1)}| = m, |D^{(2)}| = n$. Consider a cardinality-dependent coupled sampling $S(D)$: (1) $\mathcal{B}^{(1)} \sim \text{Poisson}_\gamma(D^{(1)})$ is Poisson sampling with rate $\gamma$; (2) $\mathcal{B}^{(2)} \sim \text{Sample}_{\text{WOR}}(D^{(2)}; k_{\text{neg}} \cdot |\mathcal{B}^{(1)}|)$ is sampling without replacement of size $k_{\text{neg}} \cdot |\mathcal{B}^{(1)}|$. Assume function $f$ satisfies $\left\| f(\mathcal{B}^{(1)}, \mathcal{B}^{(2)}) - f(\mathcal{B}^{(1),\prime}, \mathcal{B}^{(2),\prime}) \right\| \leq C$ for any $\mathcal{B}^{(1)} \sim_{\Delta,K} \mathcal{B}^{(1),\prime}$ and $\mathcal{B}^{(2)} \sim_r \mathcal{B}^{(2),\prime}$. Then for any $\alpha \geq 1$, Gaussian mechanism $f(S(D)) + \mathcal{N}(0, \sigma^2 C^2 I)$ is $(\alpha, \varepsilon(\alpha))$-RDP with*

$$\varepsilon(\alpha) = \frac{1}{\alpha - 1} \log \mathbb{E}_{\ell \sim \text{Bin}(m,\gamma)} \Psi_\alpha \left( (1 - \Gamma_\ell) \cdot \mathcal{N}(0, \sigma^2) + \Gamma_\ell \cdot \mathcal{N}(1, \sigma^2) \| \mathcal{N}(0, \sigma^2) \right), \quad (3)$$

*where $\text{Bin}(m, \gamma)$ stands for the Binomial distribution, effective sampling rate $\Gamma_\ell := 1 - (1-\gamma)^K (1 - \frac{\ell \cdot k_{\text{neg}}}{n})$ and $\Psi_\alpha(P\|Q) := \mathbb{E}_{x \sim Q}(P(x)/Q(x))^\alpha$ for two distributions $P, Q$.*

The proof is deferred to Appendix B. $\Psi_\alpha$ here is a univariate integral and can be numerically calculated up to arbitrary precision [32]. Although Theorem 4.1 adopts $S^{(1)}$ as Poisson sampling and $S^{(2)}$ as sampling without replacement, the analysis and results can be extended to other combinations of sampling strategies. We leave these generalizations to the extended version of this work.

So far we have addressed the two key challenges: a) *sensitivity*: the mini-batch gradient sensitivity can be bounded by a constant (Proposition 4.1), by adopting adaptive clipping (Algorithm 1) and restricting $|\mathcal{B}_-(u)| \leq 1$ (achieved by Algorithm 2); b) *coupled sampling*: the mini-batch sampling can be simplified to cardinality-dependent sampling (also achieved by Algorithm 2), and the privacy cost can be computed by Eq. (3). Integrating these ideas, we propose a DP-SGD for relational learning that leverage node sampling without replacement for negative sampling and adaptive gradient

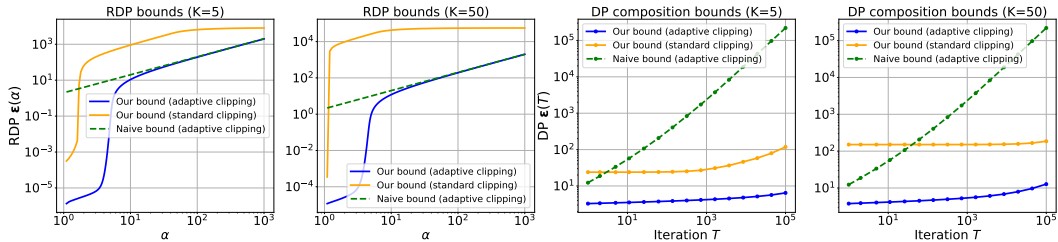

Figure 2: Comparison of per-iteration RDP bound $\varepsilon(\alpha)$ (left two figures) and DP composition bound $\varepsilon_{\mathrm{DP}}(T)$ over $T$ iterations (right two figures), under different capped node degree $K$. Our bound (adaptive clipping) refers bound Eq.(3). Our bound (standard clipping) uses similar amplification analysis but with standard clipping. Naive bound is the Gaussian RDP bound $\varepsilon(\alpha) = \alpha/\sigma^2$.

clipping, as described in Algorithm 3. The assumption in Theorem 4.1 that $f$ (i.e., gradients) has a constant sensitivity $C$ regardless of the number of removals is guaranteed by adaptive clipping and $|\mathcal{B}_-(u)| \leq 1$. As a result, Proposition 4.1 and Theorem 4.1 directly implies Corollary 4.1.

**Corollary 4.1.** *Algorithm 3 achieves $(\alpha, \varepsilon(\alpha))$-RDP with $\varepsilon(\alpha)$ defined in Eq.(3), where $D^{(1)} = \mathcal{E}, D^{(2)} = \mathcal{V}$, and $K$ is the maximal node degree.*

Note that the bound Eq.(3) depends on the maximal node degree $K$. To mitigate the influence of high-degree nodes, in practice we cap each node's degree at a desired constant $K$ by randomly sampling its neighbors, same as Algorithm 1 of [42]. The effect of degree capping on utility is explored in Figure 3 in our later experiments.

## 5 Experimental Results

In this section, we empirically evaluate the privacy and utility characteristics of our proposed method. First, we numerically compute and compare the privacy bounds (Eq.(3)). Second, we consider an application of finetuning text encoders on relational data to evaluate the privacy-utility trade-offs. In particular, the experiments demonstrate the superiority of our method over standard DP-SGD.

### 5.1 Numerical Results for Eq. (3)

**Parameters.** We set the parameters based on the real-world graph statistics (Appendix E, Table 2) we will use in the latter experiments. By default: number of nodes $n = 10^6$, number of edges $m = 5 \times 10^6$, capped node degree $K = 5$, sampling rate $\gamma = 10^{-5}$, Gaussian noise levels $\sigma = 0.5$ and number of negative edges per positive edge $k_{\mathrm{neg}} = 4$. The clipping threshold $C$ is set to $1$. We also evaluate different combinations of parameters, which can be found at Appendix D.

**Per-iteration RDP Bounds.** We evaluate the RDP bound $\varepsilon(\alpha)$ (Eq.(3)), shown in the left two subfigures of Figure 2, which characterizes the per-iteration privacy loss in DP-SGD. Our bound is compared against two baselines: (a) the naive Gaussian mechanism bound $\varepsilon_{\mathrm{Gaussian}}(\alpha) = \alpha/2\sigma^2$ [56] without any amplification analyses; (b) the amplification bound with standard gradient clipping in place of the proposed adaptive clipping. The latter is also a coupled-sampling amplification bound we derived using an analysis similar to that of Theorem 4.1, and is provided in Appendix C.

**DP Bounds over Composition.** We also compare the accumulated privacy loss of applying multiple iterations of DP-SGD in terms of $(\varepsilon, \delta)$-DP, tracked by composition theorems of RDP [56] and translating RDP to DP [75]. We choose privacy budget $\delta = 1/m = 0.2 \times 10^{-6}$ and compare $\varepsilon_{\mathrm{DP}}(T)$ as a function of iteration $T$. The results of DP bound over composition are shown in the right two subfigures in Figure 2.

We observe that the composite DP bound of our method is remarkably smaller than the naive Gaussian one as well as the amplification one with standard clipping. This strongly suggests a better model utility our proposed method can bring, which we explore in the next section.

### 5.2 Private Relational Learning for Pretrained Text Encoders

We study the application of private relational learning for finetuning a text encoder on where relational information is injected into the encoder. We use relation prediction on text-attributed graphs as the

evaluation task. The fine-tuned text encoder can be used to predict relations for unseen node sets, analogous to zero-shot link recommendation. Specifically, the training graph is treated as private, and the encoder weights are privately fine-tuned using Algorithm 3. After fine-tuning, model utility is evaluated by the relation prediction performance on a separate test graph. Note that although we adopt fine-tuning text encoders as a specific application, our algorithm can be applied to more general relational learning scenarios (e.g., to train the weights of MLPs when entities hold dense features).

Table 1: Results on relation prediction with **entity-level** differentially private relational learning. MAG(CHN→USA) means the model is fine-tuned on MAG-CHN and then tested on MAG-USA.

| Privacy | Pretrained Models | MAG(CHN→USA) PREC@1 | MRR | MAG(USA→CHN) PREC@1 | MRR | AMAZ(Sports→Cloth) PREC@1 | MRR | AMAZ(Cloth→Sports) PREC@1 | MRR |
|---|---|---|---|---|---|---|---|---|---|
| base model (w/o fine-tuning) | BERT.base | 4.41 | 9.94 | 6.48 | 12.69 | 14.90 | 22.41 | 8.36 | 14.04 |
| | BERT.large | 2.00 | 5.48 | 2.71 | 6.39 | 5.72 | 10.11 | 3.78 | 7.37 |
| | SciBERT | 8.70 | 17.12 | 13.89 | 23.96 | - | - | - | - |
| | LinkBERT.large | 1.09 | 4.01 | 1.46 | 4.75 | 4.01 | 8.60 | 2.06 | 5.37 |
| | Llama2-7B | 4.24 | 8.68 | 5.21 | 9.71 | 19.45 | 27.41 | 6.13 | 10.11 |
| $\epsilon = \infty$ (non-private fine-tuning) | BERT.base | 28.07 | 39.11 | 41.93 | 53.91 | 36.13 | 47.07 | 29.84 | 39.61 |
| | BERT.large | 26.37 | 37.73 | 40.90 | 53.16 | 36.89 | 47.50 | 29.30 | 39.76 |
| | Llama2-7B | 32.80 | 46.67 | 45.65 | 58.59 | 41.01 | 52.39 | 29.21 | 41.44 |
| $\epsilon = 10$ (standard clipping) | BERT.base | 12.44 | 21.80 | 28.26 | 40.29 | 23.66 | 33.31 | 20.18 | 29.59 |
| | BERT.large | 11.36 | 20.67 | 28.15 | 40.57 | 12.39 | 19.91 | 21.33 | 31.10 |
| | Llama2-7B | 11.90 | 21.35 | 20.10 | 32.50 | 29.99 | 41.04 | 18.14 | 27.46 |
| $\epsilon = 10$ (Ours) | BERT.base | 17.39 | 27.51 | 29.93 | 42.01 | 27.76 | 37.80 | 22.80 | 32.71 |
| | BERT.large | 18.43 | 29.36 | 31.25 | 43.83 | 19.07 | 28.17 | 24.88 | 35.01 |
| | Llama2-7B | 18.23 | 30.27 | 26.15 | 39.80 | 34.82 | 46.43 | 22.96 | 33.46 |
| $\epsilon = 4$ (standard clipping) | BERT.base | 11.03 | 20.05 | 26.69 | 38.53 | 21.24 | 30.63 | 18.31 | 27.10 |
| | BERT.large | 8.85 | 17.25 | 22.42 | 32.20 | 7.33 | 13.01 | 18.64 | 27.63 |
| | Llama2-7B | 9.73 | 17.84 | 17.96 | 29.46 | 28.00 | 38.49 | 13.86 | 21.56 |
| $\epsilon = 4$ (Ours) | BERT.base | 15.18 | 25.07 | 28.00 | 39.86 | 26.13 | 36.03 | 21.88 | 31.52 |
| | BERT.large | 15.32 | 25.78 | 28.22 | 40.78 | 16.09 | 24.57 | 23.72 | 33.76 |
| | Llama2-7B | 15.69 | 26.93 | 23.66 | 36.89 | 32.69 | 44.04 | 21.20 | 31.02 |

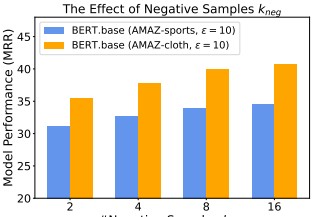 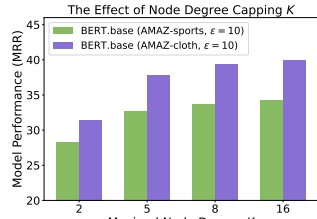 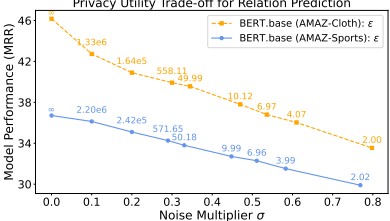

Figure 3: Utility of different #negative sample per positive $k_{\text{neg}}$, degree capping $K$, and noise multiplier $\sigma$ for zero-shot relation prediction. The legend AMAZ-sports means training on AMAZ-cloth and testing on AMAZ-sports, and similar for the legend AMAZ-cloth. In the rightmost figure, numbers along each line indicate the corresponding privacy parameter $\varepsilon_{\text{DP}}$ at different noise levels.

**Experimental Settings.** We adopt two pre-trained language models BERT [76] and Llama2 [77] as the text encoders, and four text-attributed graphs from two subdomain pairs: two citation networks (MAG-CHN, MAG-USA) [78] and two co-purchase networks (AMZA-Sports, AMZA-Cloth) [79]. The models are first privately fine-tuned on one network (e.g., MAG-CHN), and then its utility is evaluated on another same-domain network (e.g., MAG-USA). We report the overall privacy loss in terms of $(\varepsilon, \delta)$-DP, where $\delta$ is set to $1/|\mathcal{E}_{\text{train}}|$, the size of the relation set used for training after degree capping. We use the ranking metrics of top@1 precision (PREC@1) and mean reciprocal rank (MRR) to evaluate the relation prediction performance. The maximal node degree is capped by $K = 5$ according to Algorithm 3. See Appendix E for more implementation details.

**Baselines.** We compare against the following baselines: (a) Base models without fine-tuning: text encoders are applied directly to the test graph without any relational learning on the training graph and thus preserve the privacy; (b) Non-private fine-tuning: text encoders are fine-tuned on the training graph without any privacy constraints; and (c) standard clipping: text encoders are fine-tuned using Algorithm 3, but with standard per-sample gradient clipping in place of our proposed method.

**Evaluation Results.** Table 1 shows that all models fined-tuned by our proposed algorithm outperform their non-fine-tuned base models on all four corresponding test domains for relation prediction by a large margin, even under the constraints of node-level DP $\epsilon \in \{4, 10\}$. Meanwhile, it also consistently outperforms the one with standard gradient clipping under the same privacy budgets. This results validates the effectiveness of our proposed privacy-preserving algorithm for relational learning.

**Ablation study.** Figure 3 shows an ablation study of our method, examining the effect of the number of negative samples per positive $k_{\text{neg}}$, the capped node degree $K$, and privacy–utility trade-offs. We find that increasing $K, k_{\text{neg}}$ generally brings better model utility under the same privacy budget.

## 6 Conclusion

We propose a DP-SGD variant tailored for relational learning with rigorous entity-level privacy guarantees. Gradient sensitivity is tightly controlled via a carefully chosen negative sampling strategy and novel adaptive clipping. We extend privacy amplification analysis to account for mini-batches with positive and negative samples drawn from distinct yet dependent sampling mechanisms. Experiments on real-world relational data show that our method achieves better utility-privacy trade-offs than existing approaches.

## Acknowledgments and Disclosure of Funding

The work is supported by NSF awards PHY-2117997, IIS-2239565, CCF-2402816, IIS-2428777, JPMC faculty award and Meta award.

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

# A  Proof of Proposition 4.1

*Proof.* Recall in Section 4.1, we derive the sensitivity of removing node $u^*$ for neighboring mini-batches $\mathcal{B} \sim \mathcal{B}'$:

$$\|\mathbf{g}(\mathcal{B}) - \mathbf{g}(\mathcal{B}')\| \leq \sum_{T_i \in \mathcal{B}_+(u^*)} \|\mathbf{g}(T_i)\| + \sum_{T_i \in \mathcal{B}_-(u^*)} \|\mathbf{g}(T_i)\| + \sum_{T_i' \in \mathcal{B}_-'(u^*)} \|\mathbf{g}(T_i')\|. \tag{4}$$

Now consider the adaptive gradient clipping $\bar{\mathbf{g}}(T_i) := \mathbf{g}(T_i)/\max\{1, \|\mathbf{g}(T_i)/C(T_i, \mathcal{B})\|\}$ with $C(T_i, \mathcal{B}) = C/(\sup_{u \in T_i} |\mathcal{B}_+(u)| + |\mathcal{B}_-(u)|)$. For $T_i \in \mathcal{B}_+(u^*)$ or $T_i \in \mathcal{B}_-(u^*)$, since $u^* \in T_i$, we have

$$\|\bar{\mathbf{g}}(T_i)\| \leq \frac{C}{(\sup_{u \in T_i} |\mathcal{B}_+(u)| + |\mathcal{B}_-(u)|)} \leq \frac{C}{(|\mathcal{B}_+(u^*)| + |\mathcal{B}_-(u^*)|)}. \tag{5}$$

For $T_i' \in \mathcal{B}_-'(u^*)$, the node $u^*$ is replaced by arbitrary other nodes, so generally $u^* \notin T_i'$. We can only use pessimistic bound $\|\mathbf{g}(T_i')\| \leq C$. Overall, the sensivity becomes

$$\|\bar{\mathbf{g}}(\mathcal{B}) - \bar{\mathbf{g}}(\mathcal{B}')\|$$
$$\leq (|\mathcal{B}_+(u^*)| + |\mathcal{B}_-(u^*)|) \cdot \frac{C}{(|\mathcal{B}_+(u^*)| + |\mathcal{B}_-(u^*)|)} + |\mathcal{B}_-'(u^*)| \cdot C = (1 + |\mathcal{B}_-(u^*)|) \cdot C, \tag{6}$$

where we use the fact $|\mathcal{B}_-'(u^*)| = |\mathcal{B}_-(u^*)|$.

$\square$

# B  Proof of Theorem 4.1

**Theorem 4.1.** *Consider a composite dataset $D = (D^{(1)}, D^{(2)})$ of size $|D^{(1)}| = m, |D^{(2)}| = n$. Consider a cardinality-dependent coupled sampling $S(D)$: (1) $\mathcal{B}^{(1)} \sim \mathrm{Poisson}_\gamma(D^{(1)})$ is Poisson sampling with rate $\gamma$; (2) $\mathcal{B}^{(2)} \sim \mathrm{Sample}_{\mathrm{WOR}}(D^{(2)}; k_{\mathrm{neg}} \cdot |\mathcal{B}^{(1)}|)$ is sampling without replacement of size $k_{\mathrm{neg}} \cdot |\mathcal{B}^{(1)}|$. Assume function $f$ satisfies $\left\|f(\mathcal{B}^{(1)}, \mathcal{B}^{(2)}) - f(\mathcal{B}^{(1),'}, \mathcal{B}^{(2),'})\right\| \leq C$ for any $\mathcal{B}^{(1)} \sim_{\Delta,K} \mathcal{B}^{(1),'}$ and $\mathcal{B}^{(2)} \sim_r \mathcal{B}^{(2),'}$. Then for any $\alpha \geq 1$, Gaussian mechanism $f(S(D)) + \mathcal{N}(0, \sigma^2 C^2 I)$ is $(\alpha, \varepsilon(\alpha))$-RDP with*

$$\varepsilon(\alpha) = \frac{1}{\alpha - 1} \log \mathbb{E}_{\ell \sim \mathrm{Bin}(m,\gamma)} \Psi_\alpha \left((1 - \Gamma_\ell) \cdot \mathcal{N}(0, \sigma^2) + \Gamma_\ell \cdot \mathcal{N}(1, \sigma^2) \| \mathcal{N}(0, \sigma^2)\right), \tag{3}$$

*where $\mathrm{Bin}(m, \gamma)$ stands for the Binomial distribution, effective sampling rate $\Gamma_\ell := 1 - (1 - \gamma)^K (1 - \frac{\ell \cdot k_{\mathrm{neg}}}{n})$ and $\Psi_\alpha(P\|Q) := \mathbb{E}_{x \sim Q}(P(x)/Q(x))^\alpha$ for two distributions $P, Q$.*

*Proof.* **Notation.** We use the removal/insertion notation of DP: we have two neighboring graphs $G = (\mathcal{V}, \mathcal{E}), G' = (\mathcal{V}', \mathcal{E}')$ where $\mathcal{V} = \{1, 2, ..., n\}$, $\mathcal{V}' = \{1, 2, ..., n-1\}$ and $\mathcal{E} = \mathcal{E}' \cup \{(n, u_i) : i = 1, 2, ..., K\}$. Let $m = |\mathcal{E}|$. Let $\mathbb{X} = 2^\mathcal{E}, \mathbb{X}' = 2^{\mathcal{E}'}$ be the all possible subset of edges that represents the space of positive samplings. Let $\mathbb{Y} = \{(y_1, y_2..., y_{b \cdot k_{\mathrm{neg}}}) : y_i \in V, b \in [m], y_i \neq y_j\}$ represent all possible ordered node set that represents the space of negative samplings. For simplicity, we will assume $k_{\mathrm{neg}} = 1$, i.e., each positive sample is only paired with one negative sample. In the end we will show how to adapt to case $k_{\mathrm{neg}} > 1$. An actual dataset we are computing on is $(x, y)$ for some $x \in \mathbb{X}, y \in \mathbb{Y}$ or $(x', y')$ for some $x' \in \mathbb{X}', y \in \mathbb{Y}'$ (again, $\mathbb{X}', \mathbb{Y}'$ are the sampling space of positive edges and ordered node set defined on neighboring graph $\mathcal{G}'$). We denote $\mu_{x,y}$ as the output distribution of the mechanism acting on a fixed sampled dataset $(x, y)$, and denote $p, q$ as the mixture output distribution of the mechanism acting on a randomly sampled dataset $(x, y)$ and $(x', y')$ respectively. By definition, we have

$$p = \sum_{x,y} \omega(x, y) \mu_{x,y}, \quad q = \sum_{x',y'} \omega'(x', y') \mu_{x',y'}, \tag{7}$$

where $\omega(x, y), \omega'(x', y')$ are probabilistic distributions of the samplings. With an abuse of notation, we also write $\omega(x, y) = \omega(x)\omega(y|x)$ and $\omega'(x', y') = \omega'(x')\omega'(y'|x')$. Our goal is to bound $\Psi_\alpha(p\|q) = \mathbb{E}_q(p/q)^\alpha$ and $\Psi_\alpha(q\|p) = \mathbb{E}_p(q/p)^\alpha$, i.e., the logarithm of Rényi divergence.

**Proof idea.** The proof is basically divided into three steps: (1) we will show how it is possible to extend Conditional Coupling [54] to this product data $(x, y)$; (2) a concrete conditional coupling is constructed and proved to be valid. Using the conditional coupling, we can apply Jensen's inequality to obtain a bound for RDP $\varepsilon$; (3) finally, we derive the final bound by considering the properties of specific Algorithm 3, i.e., a constant-sensitivity mechanism.

**Jensen's inequality by conditional coupling.** The key idea is to apply Jensen's inequality to $\Psi_\alpha(\cdot||\cdot)$ as $\Psi_\alpha$ is jointly convex [80]. To do so, we need to re-write $p, q$ in terms of balancing summation. we will generalize the conditional coupling technique [54] to our case. In the following we will show how we can achieve this in a high-level view. Consider the conditional distributions $\omega(x|A_i)$, where $A_i$ represents the event "$x$ has $i$ removed edges (i.e., the edges involve the removed node $n$)", and conditional distributions $\omega(y|B_j, x)$, $j = 0, 1$, where $B_0(B_1)$ represents the event "$[y]_{S(x)}$ has (no) removed node $n$", where $S(x)$ is a subset of indices defined by $S(x) \equiv \{\ell \in \{1, 2, ..., |x|\} : x_\ell$ has no removed edges$\}$. We define coupling $\pi$ as a probabilistic measure on $\mathbb{X}^{K+1} \times \mathbb{X}' \times \mathbb{Y}^{2(K+1)} \times \mathbb{Y}'$. As a shorthand we write $\vec{x} = (x_0, ..., x_K) \in \mathbb{X}^{K+1}$, $\vec{y}^{(j)} = (y_0^{(j)}, ..., y_K^{(j)}) \in \mathbb{Y}^{K+1}$ and $\vec{y} = (\vec{y}^{(0)}, \vec{y}^{(1)})$. We require coupling $\pi(\vec{x}, x', \vec{y}, y') = \pi(\vec{x}, x')\pi(\vec{y}, y'|\vec{x}, x')$ to satisfy:

$$\sum_{\vec{x} \backslash \{x_i\}, x'} \pi(\vec{x}, x') = \omega(x_i|A_i), \tag{8}$$

$$\sum_{\vec{x}} \pi(\vec{x}, x') = \omega'(x'), \tag{9}$$

$$\sum_{\vec{y} \backslash \{y_i^{(j)}\}, y'} \pi(\vec{y}, y'|\vec{x}, x') = \omega(y_i^{(j)}|B_j, x_i), \tag{10}$$

$$\sum_{\vec{y}} \pi(\vec{y}, y'|\vec{x}, x') = \omega'(y'|x'). \tag{11}$$

Specifically, let us denote the size $|S(x_i)|$ by $L$ and define $p_{n,m} \equiv 1/n(n-1)\cdots(n-m+1)$ be the inverse of falling factorial from $n$ to $n - m + 1$, then we can explicitly write down the marginal probability:

$$\omega(x_i|A_i) = \frac{\gamma^{|x_i|}(1-\gamma)^{m-|x_i|}}{\binom{K}{i}\gamma^i(1-\gamma)^{K-i}} = \frac{1}{\binom{K}{i}}\gamma^{|x_i|-i}(1-\gamma)^{m-K-|x_i|+i}, \tag{12}$$

$$\omega'(x') = \gamma^{|x'|}(1-\gamma)^{m-K-|x'|}, \tag{13}$$

$$\omega(y_i^{(0)}|B_0, x_i) = p_{n-1,L}p_{n-L,|x_i|-L}, \tag{14}$$

$$\omega(y_i^{(1)}|B_1, x_i) = \frac{1}{L}p_{n-1,L-1}p_{n-L,|x_i|-L} = \frac{1}{L}p_{n-1,|x_i|-1}, \tag{15}$$

$$\omega'(y') = p_{n-1,|x'|}. \tag{16}$$

Suppose these requirements are satisfied, then one can rewrite $p, q$ in terms of the coupling distribution $\pi(\vec{x}, x', \vec{y}, y')$. We start with rewriting $p$:

$$p = \sum_{x,y} \omega(x,y)\mu_{x,y} = \sum_{x,y} \omega(x)\omega(y|x)\mu_{x,y} \tag{17}$$

$$= \sum_{x,y} \left(\sum_i \omega(x|A_i)\omega(A_i)\right) \left(\sum_j \omega(y|x, B_j)\omega(B_j|A)\right) \mu_{x,y} \tag{18}$$

$$= \sum_{i,j} \sum_{x,y} \omega(x|A_i)\omega(A_i)\omega(y|x, B_j)\omega(B_j|x)\mu_{x,y} \tag{19}$$

Now we for each index $i, j$, we can rename the dummy variable $x, y$ by $x_i, y_i^{(j)}$,

$$\sum_{i,j} \sum_{x,y} \omega(x|A_i)\omega(A_i)\omega(y|x, B_j)\omega(B_j|x)\mu_{x,y} \tag{20}$$

$$= \sum_{i,j} \sum_{x_i, y_i^{(j)}} \omega(x_i|A_i)\omega(y_i^{(j)}|x_i, B_j)\omega(A_i)\omega(B_j|x_i)\mu_{x_i, y_i^{(j)}} \tag{21}$$

$$= \sum_{i,j} \sum_{x_i, y_i^{(j)}} \left( \sum_{\vec{x}\setminus\{x_i\}, x'} \pi(\vec{x}, x') \right) \left( \sum_{\vec{y}\setminus\{y_i^{(j)}\}, y'} \pi(\vec{y}, y'|\vec{x}, x') \right) \omega(A_i)\omega(B_j|x_i)\mu_{x_i, y_i^{(j)}} \tag{22}$$

$$= \sum_{i,j} \sum_{\vec{x}, x', \vec{y}, y'} \pi(\vec{x}, x')\pi(\vec{y}, y'|\vec{x}, x')\omega(A_i)\omega(B_j|x_i)\mu_{x_i, y_i^{(j)}} \tag{23}$$

$$= \sum_{\vec{x}, x', \vec{y}, y'} \pi(\vec{x}, x', \vec{y}, y') \sum_{i,j} \omega(A_i)\omega(B_j|x_i)\mu_{x_i, y_i^{(j)}}. \tag{24}$$

Similarly, we can rewrite $q$ in terms of $\pi$:

$$q = \sum_{x', y'} \omega'(x', y')\mu_{x', y'} = \sum_{x', y'} \omega'(x')\omega(y'|x')\mu_{x', y'} \tag{25}$$

$$= \sum_{x', y'} \left( \sum_{\vec{x}} \pi(\vec{x}, x') \right) \left( \sum_{\vec{y}} \pi(\vec{y}, y'|\vec{x}, x') \right) \mu_{x', y'} \tag{26}$$

$$= \sum_{\vec{x}, x', \vec{y}, y'} \pi(\vec{x}, x', \vec{y}, y')\mu_{x', y'}. \tag{27}$$

These imply that by convexity of $\Psi_\alpha(\cdot||\cdot)$, we have

$$\Psi_\alpha(p||q) \leq \sum_{\vec{x}, x', \vec{y}, y'} \pi(\vec{x}, x', \vec{y}, y')\Psi_\alpha \left( \sum_{i,j} \omega(A_i)\omega(B_j|x_i)\mu_{x_i, y_i^{(j)}} || \mu_{x', y'} \right), \tag{28}$$

$$\Psi_\alpha(q||p) \leq \sum_{\vec{x}, x', \vec{y}, y'} \pi(\vec{x}, x', \vec{y}, y')\Psi_\alpha \left( \mu_{x', y'} || \sum_{i,j} \omega(A_i)\omega(B_j|x_i)\mu_{x_i, y_i^{(j)}} \right). \tag{29}$$

**Construct coupling.** Now we are going to construct a specific $\pi$ to bound $\Psi_\alpha(p||q)$ and $\Psi_\alpha(q||p)$. The sampling process of $\pi(\vec{x}, x', \vec{y}, y')$ goes as follows:

- first sample $x_0 \sim \omega(\cdot|A_0)$, and denote the size of $x_0$ by $L$;

- for each $i = 1, 2, ..., K$, sample $x_i$ by randomly drawing a size-$i$ subset of the removed edges and adding the subset to $x_0$;

- let $x' = x_0$;

- sample $y_0^{(0)} \sim \omega(\cdot|B_0, x_0)$;

- then sample $y_0^{(1)}$ by randomly replacing one node in $y_0^{(0)}$ with the new node $n$;

- for each $i = 1, ..., K$,

    - sample $y_i^{(0)}$ by randomly choosing $i$ nodes (not in $y_0^{(0)}$) to and adding them to $y_0^{(0)}$;
    - sample $y_i^{(1)}$ by first randomly replacing one node in $y_0^{(0)}$ with the removed node $n$ (call this intermediate sample $\tilde{y}$) and then randomly choosing $i$ nodes (not in $\tilde{y}$ and not include the new node $n$) and adding to $\tilde{y}$.

- let $y' = y_0^{(0)}$.

Note that this particularly construction yields the following factorization:

$$\pi(\vec{x}, x', \vec{y}, y') = \pi(\vec{x}, x')\pi(\vec{y}, y'|\vec{x}, x'), \tag{30}$$

$$\pi(\vec{x}, x') = \pi(x_0)\pi(x'|x_0)\prod_{i=1}^{K}\pi(x_i|x_0), \tag{31}$$

$$\pi(\vec{y}, y'|\vec{x}, x') = \pi(y_0^{(0)})\pi(y'|y_0^{(0)})\pi(y_0^{(1)}|y_0^{(0)})\prod_{i=1}^{K}\pi(y_i^{(0)}|y_0^{(0)})\pi(y_i^{(1)}|y_0^{(0)}). \tag{32}$$

Now let's verify the coupling constraints Eq.(8) to (11). For Eq.(8), when $i = 0$, we have $\sum_{\vec{x}\setminus\{x_0\}, x'}\pi(\vec{x}, x') = \pi(x_0) = \omega(x_0|A_0)$ as desired. When $i \geq 1$,

$$\sum_{\vec{x}\setminus\{x_i\}, x'}\pi(\vec{x}, x') = \sum_{x_0}\pi(x_0)\pi(x_i|x_0) = \gamma^L(1-\gamma)^{m-K-L} \cdot \frac{1}{\binom{K}{i}} = \omega(x_i|A_i), \tag{33}$$

where we use the basic fact from our construction that $\pi(x_0) = \omega(x_0|A_0) = \gamma^L(1-\gamma)^{m-K-L}$, $\pi(x_i|x_0) = 1/\binom{K}{i}$, and $\omega(x_i|A_i) = \gamma^{L+i}(1-\gamma)^{m-L-i}/(\binom{K}{i}\gamma^i(1-\gamma)^{K-i}) = \gamma^L(1-\gamma)^{m-K-L}/\binom{K}{i}$. Thus Eqn.(8) holds. For Eqn.(9), clearly $\sum_{\vec{x}}\pi(\vec{x}, x') = \sum_{x_0}\pi(x_0)\pi(x'|x_0) = \sum_{x_0}\pi(x_0)\delta(x' - x_0) = \pi(x') = \omega(x'|A_0) = \gamma^L(1-\gamma)^{m-K-L} = \omega'(x')$. So Eqn.(9) holds as well.

For Eqn.(10), when $i = j = 0$, we have $\sum_{\vec{y}\setminus\{y_0^{(0)}\}, y'}\pi(\vec{y}, y'|\vec{x}, x') = \pi(y_0^{(0)}) = \omega(y_0^{(0)}|B_0, x_0)$ by our construction. When $i \neq 0, j = 0$, recall that $p_{n,m} \equiv 1/n(n-1)\cdots(n-m+1)$ is the inverse of falling factorial from $n$ to $n - m + 1$, then

$$\sum_{\vec{y}\setminus\{y_i^{(0)}\}, y'}\pi(\vec{y}, y'|\vec{x}, x') = \sum_{y_0^{(0)}}\pi(y_0^{(0)}|x_0)\pi(y_i^{(0)}|y_0^{(0)}, x_i) = p_{n-1,L}p_{n-L,i} = p_{n-1,L}p_{n-L,|x_i|-L}$$
$$= \omega(y_i^{(0)}|B_0, x_i). \tag{34}$$

Here we use the fact that $|x_i| = |x_0| + i = L + i$. When $i \neq 0, j = 1$, we have

$$\sum_{\vec{y}\setminus\{y_i^{(1)}\}, y'}\pi(\vec{y}, y'|\vec{x}, x') = \sum_{y_0^{(0)}}\pi(y_0^{(0)}|x_0)\pi(y_i^{(1)}|y_0^{(0)}, x_i) = (n - L) \cdot p_{n-1,L}\frac{1}{L}p_{n-L,i}$$
$$= \frac{1}{L}p_{n-1,L+i-1} = \frac{1}{L}p_{n-1,|x_i|-1} = \omega(y_i^{(1)}|B_1, x_i). \tag{35}$$

Finally, $\sum_{\vec{y}}\pi(\vec{y}, y'|\vec{x}, x') = \omega(y'|A_0, x_0) = p_{n-1,L} = p_{n-1,|x'|}$. Therefore, the constructed $\pi$ is a valid coupling and thus Eqns(28), (29) hold.

**Derive explicit bound.** Now we are going to derive bounds for Eqns.(28) and (29) by plugging $\pi$ into it. Note that coefficients $\{\omega(A_i)\omega(B_j|x_i)\}_{i,j}$ satisfy $\sum_{i,j}\omega(A_i)\omega(B_j|x_i) = 1$ and thus $\sum_{i,j\neq 0}\omega(A_i)\omega(B_j|x_i) = 1 - \omega(A_0)\omega(B_0|x_0)$. Therefore, we can write

$$\sum_{i,j}\omega(A_i)\omega(B_j|x_i)\mu_{x_i,y_i^{(j)}} = \omega(A_0)\omega(B_0|x_0)\mu_{x_0,y_0^{(0)}} + \sum_{i,j\neq 0}\omega(A_i)\omega(B_j|x_i)\mu_{x_i,y_i^{(j)}} \tag{36}$$

$$= \sum_{i,j\neq 0}\frac{\omega(A_i)\omega(B_j|x_i)}{1 - \omega(A_0)\omega(B_0|x_0)}\left(\omega(A_0)\omega(B_0|x_0) \cdot \mu_{x_0,y_0^{(0)}} + (1 - \omega(A_0)\omega(B_0|x_0)) \cdot \mu_{x_i,y_i^{(j)}}\right). \tag{37}$$

Notice that $\{\omega(A_i)\omega(B_j|x_i)/(1 - \omega(A_0)\omega(B_0|x_0))\}_{i,j\neq 0}$ are coefficients of convex combination. Thus by Jensen's inequality, Eqn.(28) becomes

$$\Psi_\alpha(p\|q) \leq \sum_{\vec{x}, x', \vec{y}, y', i, j\neq 0}\pi(\vec{x}, x', \vec{y}, y')\frac{\omega(A_i)\omega(B_j|x_i)}{1 - \omega(A_0)\omega(B_0|x_0)}$$
$$\cdot \Psi_\alpha\left(\omega(A_0)\omega(B_0|x_0) \cdot \mu_{x_0,y_0^{(0)}} + (1 - \omega(A_0)\omega(B_0|x_0)) \cdot \mu_{x_i,y_i^{(j)}}\|\mu_{x',y'}\right), \tag{38}$$

and similarly for $\Psi_\alpha(q||p)$. It is worthy noticing that this Jensen's inequality is generally not tight. Fortunately, if we assume the sensitivity of Gaussian mechanism $\mu_{x_i, y_i^{(j)}}$ is a constant regardless of number of removals $i$ and replacements $j$, then the Jensen'inequality is indeed tight.

The coupling $\pi$ ensures that $x_0 = x'$, $y_0^{(0)} = y'$ and $(x_i, y_i^{(j)})$ is "neighboring" to $(x_0, y_0^{(0)})$, i.e., the difference of the mean of $\mu_{x_0, y_0^{(0)}}$ and of $\mu_{x_i, y_i^{(j)}}$ is bounded. On the other hand, the coefficient $\omega(B_0|x_0) = 1 - |x_0|/n$ depends on the size of $x_0$, which is Binomial distributed $|x_0| \sim \text{Bin}(m, \gamma)$. That means the divergence $\Psi_\alpha(\cdot||\cdot)$ varies with different $|x_0|$. To reflect this dependency, we write the sum conditioning on different $|x_0| = \ell$:

$$\Psi_\alpha(p||q) \le$$
$$\sum_\ell \text{Bin}(\ell|m, \gamma) \sum_{\vec{x}, x', \vec{y}, y', i, j \ne 0} \pi(\vec{x}, x', \vec{y}, y'||x_0| = \ell) \frac{\omega(A_i)\omega(B_j|x_i)}{1 - \omega(A_0)\omega(B_0|x_0)} \tag{39}$$
$$\cdot \Psi_\alpha \left( \omega(A_0)\omega(B_0|x_0) \cdot \mu_{x_0, y_0^{(0)}} + (1 - \omega(A_0)\omega(B_0|x_0)) \cdot \mu_{x_i, y_i^{(j)}} || \mu_{x', y'} \right).$$

Now to get rid of the coefficients $\pi$, we take supremum over all possible neighboring $x_0 \sim x_i$ and $y_0^{(0)} \sim y_i^{(j)}$, subject to $|x_0| = \ell$. That is, we are interested in

$$A_\alpha(i, j, \ell) \equiv \sup_{\vec{x}, \vec{y}, |x_0| = \ell} \Psi_\alpha \left( \omega(A_0)\omega(B_0|x_0) \cdot \mu_{x_0, y_0^{(0)}} + (1 - \omega(A_0)\omega(B_0|x_0)) \cdot \mu_{x_i, y_i^{(j)}} || \mu_{x_0, y_0^{(0)}} \right).$$
$$\tag{40}$$

Since $\mu_{x_i, y_i^{(j)}}$ is isotropic Gaussian, $\Psi_\alpha(\cdot||\cdot)$ is rotational invariant, and the supremum reduces to univariate Gaussian:

$$A_\alpha(i, j, \ell)$$
$$= \Psi_\alpha \left( \omega(A_0)\omega(B_0|x_0) \cdot N(0, \sigma^2) + (1 - \omega(A_0)\omega(B_0|x_0)) \cdot N(L_2(i, j, \ell), \sigma^2) || N(0, \sigma^2) \right),$$
$$\tag{41}$$

where $L_2(i, j, \ell)$ is the L2-sensitivity of neighboring $x_0 \sim x_i, y_0^{(0)} \sim y_i^{(j)}$ under $|x_0| = \ell$. Note that Theorem 4.1 assumes $L_2$ is a constant $C$ (WLGO we further take $L_2 = C = 1$) regardless of $i, j, \ell$ if we use adaptive clipping. As a result, $A_\alpha(i, j, \ell) = A_\alpha(\ell)$ is only a function of $\ell$ (as $\omega(B_0|x_0) = 1 - \ell/n$), and

$$\Psi_\alpha(p||q) \le \sum_\ell \text{Bin}(\ell|m, \gamma) \sum_{\vec{x}, x', \vec{y}, y', i, j \ne 0} \pi(\vec{x}, x', \vec{y}, y'||x_0| = \ell) \frac{\omega(A_i)\omega(B_j|x_i)}{1 - \omega(A_0)\omega(B_0|x_0)} A_\alpha(\ell)$$
$$= \sum_\ell \text{Bin}(\ell|m, \gamma) A_\alpha(\ell). \tag{42}$$

For another side $\Psi_\alpha(q||p)$, we can follow the same procedure and use the fact $\Psi_\alpha(N(0, \sigma^2)||(1 - q)N(0, \sigma^2) + qN(c, \sigma^2)) \le \Psi_\alpha((1-q)N(0, \sigma^2) + qN(c, \sigma^2)||N(0, \sigma^2))$ [32]. This will lead to the same upper bound $\Psi_\alpha(q||p) \le \sum_\ell \text{Bin}(\ell|m, \gamma) A_\alpha(\ell)$. Therefore, the amplified $\epsilon(\alpha)$ satisfies

$$\epsilon(\alpha) \le \frac{1}{\alpha - 1} \log \left\{ \sum_\ell \text{Bin}(\ell|m, \gamma) A_\alpha(\ell) \right\} \tag{43}$$

The only thing left is to calculate $A_\alpha(\ell)$. By defining $\Gamma_\ell \equiv 1 - \omega(A_0)\omega(B_0|x_0) = 1 - (1-\gamma)^K(1 - \ell/n)$, we can write

$$A_\alpha(\ell) = \Psi_\alpha((1 - \Gamma_\ell)\mathcal{N}(0, \sigma^2) + \Gamma_\ell\mathcal{N}(1, \sigma^2) || \mathcal{N}(0, \sigma^2)), \tag{44}$$

which is well studied in previous literature and can be numerically calculated up to any desired precision. See [32]. Finally, to generalize the case where each positive sample can be paired with $k_\text{neg} > 1$ negative samples, we can think of first copy positive samples $k_\text{neg}$ times and follow the same derivation above. It is thus equivalent to replace $\ell$ by $k_\text{neg}\ell$ in $\Gamma_\ell$, with others remaining the same. □

## C  Amplification Bound for Linear Sensitivity (Standard Clipping)

In this section, we derive privacy bounds for cardinality-dependent sampling as stated in Theorem 4.1, but with standard clipping in place of adaptive clipping. Recall that if we apply standard gradient clipping, the sensitivity is $C \cdot (|\mathcal{B}_+(u^*)| + 2 \cdot |\mathcal{B}_-(u^*)|)$ according to Eq.(2).

**Theorem C.1.** *Consider a composite dataset $D = (D^{(1)}, D^{(2)})$ of size $|D^{(1)}| = m, |D^{(2)}| = n$. Consider a cardinality-dependent coupled sampling $S(D)$: (1) $\mathcal{B}^{(1)} \sim \text{Poisson}_\gamma(D^{(1)})$ is Poisson sampling with rate $\gamma$; (2) $\mathcal{B}^{(2)} \sim \text{Sample}_{\text{WOR}}(D^{(2)}; k_{\text{neg}} \cdot |\mathcal{B}^{(1)}|)$ is sampling without replacement of size $k_{\text{neg}} \cdot |\mathcal{B}^{(1)}|$. Suppose a function $f$ satisfies*

$$\left\| f(\mathcal{B}^{(1)}, \mathcal{B}^{(2)}) - f(\mathcal{B}^{(1),\prime}, \mathcal{B}^{(2),\prime}) \right\| \leq C \cdot (k+2), \quad \text{if } \mathcal{B}^{(1)} \sim_{\Delta,k} \mathcal{B}^{(1),\prime}, \mathcal{B}^{(2)} \sim_r \mathcal{B}^{(2),\prime}, \quad (45)$$

$$\left\| f(\mathcal{B}^{(1)}, \mathcal{B}^{(2)}) - f(\mathcal{B}^{(1),\prime}, \mathcal{B}^{(2)}) \right\| \leq C \cdot k, \quad \text{if } \mathcal{B}^{(1)} \sim_{\Delta,k} \mathcal{B}^{(1),\prime} \quad (46)$$

*for any $k \leq K$. Then for any $\alpha \geq 1$, Gaussian mechanism $f(S(D)) + \mathcal{N}(0, \sigma^2 C^2 I)$ is $(\alpha, \varepsilon(\alpha))$-RDP with*

$$\varepsilon(\alpha) = \frac{1}{\alpha - 1} \max \left\{ \log \mathbb{E}_{\ell \sim \text{Bin}(m,\gamma)} \Psi_\alpha \left( \sum_{i=0}^{K} \sum_{j=0}^{1} \omega_{i,j}(\ell) \cdot \mathcal{N}(i + 2j, \sigma^2) \middle\| \mathcal{N}(0, \sigma^2) \right), \right.$$

$$\left. \log \mathbb{E}_{\ell \sim \text{Bin}(m,\gamma)} \Psi_\alpha \left( \mathcal{N}(0, \sigma^2) \middle\| \sum_{i=0}^{K} \sum_{j=0}^{1} \omega_{i,j}(\ell) \cdot \mathcal{N}(i + 2j, \sigma^2) \right) \right\} \quad (47)$$

*Here $\text{Bin}(m, \gamma)$ stands for the Binomial distribution, $\omega_{i,0}(\ell) := \binom{K}{i} \gamma^i (1-\gamma)^{K-i} (1 - \ell \cdot k_{\text{neg}}/n)$ and $\omega_{i,1}(\ell) := \binom{K}{i} \gamma^i (1-\gamma)^{K-i} \ell \cdot k_{\text{neg}}/n$, and $\Psi_\alpha(P\|Q) := \mathbb{E}_{x \sim Q}(P(x)/Q(x))^\alpha$ for two distributions $P, Q$.*

*Proof.* The proof first follows the exact same coupling construction in the proof of Theorem 4.1 in Appendix B, up to coupling bounds Equations 28 and 29:

$$\Psi_\alpha(p\|q) \leq \sum_{\vec{x}, x', \vec{y}, y'} \pi(\vec{x}, x', \vec{y}, y') \Psi_\alpha \left( \sum_{i,j} \omega(A_i) \omega(B_j | x_i) \mu_{x_i, y_i^{(j)}} \middle\| \mu_{x', y'} \right), \quad (48)$$

$$\Psi_\alpha(q\|p) \leq \sum_{\vec{x}, x', \vec{y}, y'} \pi(\vec{x}, x', \vec{y}, y') \Psi_\alpha \left( \mu_{x', y'} \middle\| \sum_{i,j} \omega(A_i) \omega(B_j | x_i) \mu_{x_i, y_i^{(j)}} \right). \quad (49)$$

Again, we first condition on $|x_0| = \ell$ and obtain (we take $\Psi_\alpha(p\|q)$ for example, similar holds for $\Psi_\alpha(q\|p)$)

$$\Psi_\alpha(p\|q) \leq \sum_{\ell} \text{Bin}(\ell|m, \gamma) \sum_{\vec{x}', x', \vec{y}, y'} \pi(\vec{x}, x', \vec{y}, y' \| x_0 = \ell) \Psi_\alpha \left( \sum_{i,j} \omega(A_i) \omega(B_j | x_i) \mu_{x_i, y_i^{(j)}} \middle\| \mu_{x', y'} \right). \quad (50)$$

Recall that in the coupling $\pi$, $\pi(\vec{x}, x', \vec{y}, y') \neq 0$ if and only if $x' = x_0, y' = y)$, $x_i$ has $i$ extra samples than $x_0$, $y_i^{(0)}$ has no removal node $n$ and $y_i^{(1)}$ has one removal node $n$. We can take the worst case and obtain

$$\Psi_\alpha(p\|q) \leq \sum_{\ell} \text{Bin}(\ell|m, \gamma) \sup_{\vec{x}, x', \vec{y}, y' : \pi(\vec{x}, x', \vec{y}, y' \| x_0 = \ell) \neq 0} \Psi_\alpha \left( \sum_{i,j} \omega(A_i) \omega(B_j | \ell) \mu_{x_i, y_i^{(j)}} \middle\| \mu_{x', y'} \right). \quad (51)$$

here note that $\omega(B_j | x_i)$ only depends on the size of $S(x_i)$, i.e., the subset of $x_i$ without the removal node $n$. We have $|S(x_i)| = |x_0| = \ell$ due to our construction of coupling $\pi$. Given we are using Gaussian mechanism, that is $\mu_{x_i, y_i^{(j)}} = \mathcal{N}(\mu(x_i, y_i^{(j)}), \sigma^2 I)$ with sensitivity $\left\| \mu(x_i, y_i^{(j)}), \mu(x_{i'}, y_{i'}^{(j')}) \right\| \leq$

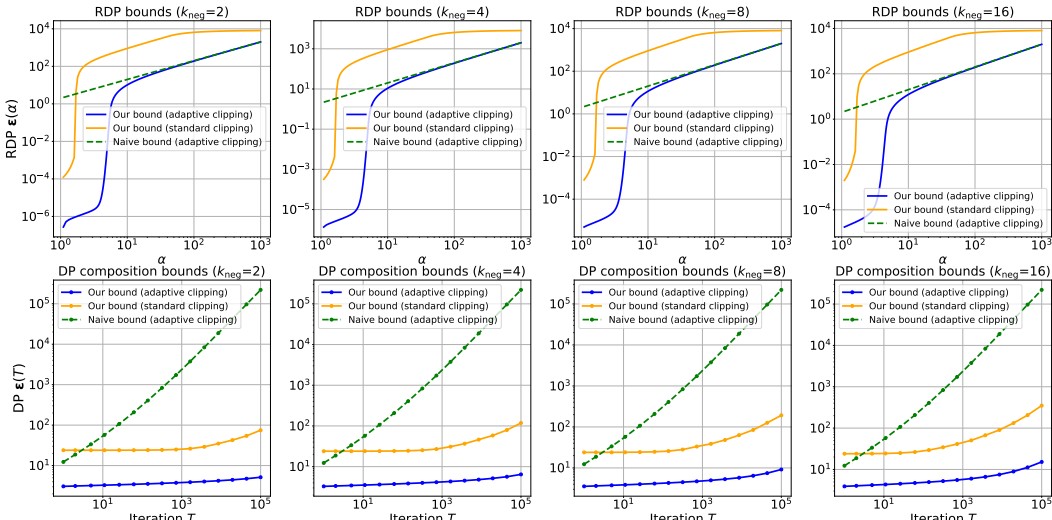

Figure 4: Comparison of per-iteration RDP bound $\varepsilon(\alpha)$ (first row) and DP composition bound $\varepsilon_{\mathrm{DP}}(T)$ over $T$ iterations (second figures), under different number of negative edges per positive $k_{\mathrm{neg}}$. Our bound (adaptive clipping) refers to the bound in Theorem 4.1. Our bound (standard clipping) refers to the similar amplification bound with standard clipping in place of adaptive clipping. Naive bound is simply the Gaussian mechanism RDP bound $\varepsilon(\alpha) = \alpha/\sigma^2$.

$|i-i'|+2\cdot|j-j'|$, the worst case is achieved by univariate Gaussian (Theorem 3.8, [54]). Therefore,

$$\Psi_\alpha(p\|q) \leq \sum_\ell \mathrm{Bin}(\ell|m,\gamma)\Psi_\alpha\left(\sum_{i,j}\omega(A_i)\omega(B_j|\ell)\cdot\mathcal{N}(i+2j,\sigma^2)\|\mathcal{N}(0,\sigma^2)\right), \tag{52}$$

$$\Psi_\alpha(q\|p) \leq \sum_\ell \mathrm{Bin}(\ell|m,\gamma)\Psi_\alpha\left(\mathcal{N}(0,\sigma^2)\|\sum_{i,j}\omega(A_i)\omega(B_j|\ell)\cdot\mathcal{N}(i+2j,\sigma^2)\right). \tag{53}$$

Finally, recall that $\omega(A_i) = \binom{K}{i}\gamma^i(1-\gamma)^{K-i}$ and $\omega(B_0|\ell) = 1 - \ell/n$ and $\omega(B_1|\ell) = \ell/n$. For $k_{\mathrm{neg}} > 1$, we can replace $\ell$ by $k_{\mathrm{neg}} \cdot \ell$, as we argued in Appendix B. This finishes the proof. $\qquad\square$

## D   Numerical Results of RDP and DP Bounds

We provide more results of comparison of per-iteration RDP and DP composite bounds over $T$ iterations (Figures 4 and 5) at different combinations of parameters. Again, by default we have: number of nodes $n = 10^6$, number of edges $m = 5 \times 10^6$, capped node degree $K = 5$, sampling rate $\gamma = 10^{-5}$, Gaussian noise levels $\sigma = 0.5$ and number of negative edges per positive edge $k_{\mathrm{neg}} = 4$. We vary one of them while keeping others unchanged. To obtain composite DP bounds, the composite RDP parameters are translated into DP parameters using the following formula.

**Theorem D.1** ([75]). *If an algorithm $M$ is $(\alpha,\varepsilon)$-RDP, then it is $(\varepsilon + \log((\alpha-1)/\alpha) - (\log\delta + \log\alpha)/(\alpha-1), \delta)$-DP for any $0 < \delta < 1$.*

## E   More Experimental Details and Results

**Dataset.** There are four text-attributed graphs among two subdomain pairs, i.e., citation relations of papers written by authors from USA and China in Microsoft Academia Graphs (MAG, [78]) and co-purchased relations in clothing and sports categories from Amazon Shopping Graphs (AMAZ, [79]). The statistics of the real-world dataset are summarized in Table 2. We followed the same procedure of preprocessing datasets and performing evaluation for all models as [81]. For degree capping, we adapted the approach from [42] to drop edges and obtain the degree-capped training set.

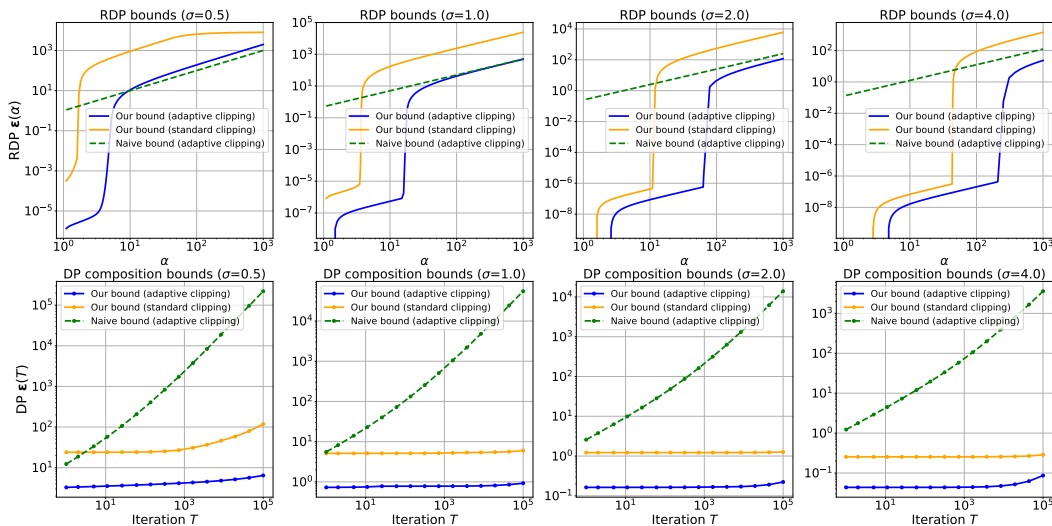

Figure 5: Comparison of per-iteration RDP bound $\varepsilon(\alpha)$ (first row) and DP composition bound $\varepsilon_{\mathrm{DP}}(T)$ over $T$ iterations (second figures), under different noise level $\sigma$. Our bound (adaptive clipping) refers to the bound in Theorem 4.1. Our bound (standard clipping) refers to the similar amplification bound with standard clipping in place of adaptive clipping. Naive bound is simply the Gaussian mechanism RDP bound $\varepsilon(\alpha) = \alpha/\sigma^2$.

**Experimental details.** Each LLM is fine-tuned through the proposed Algorithm 3 using the InfoNCE loss. The overall privacy loss is tracked through Theorem 4.1 and converted to $(\epsilon, \delta)$-DP, where $\delta$ is set to $1/|\mathcal{E}_{\mathrm{train}}|$, the size of the relation set used for training after degree capping. Note that we actually adopt Adam optimizer to update model parameters in Algorithm 3, which has the same privacy guarantee of SGD-style update in the original form, due to the post-processing property of DP [82, 26].

**Compute Resources.** We use a server with two AMD EPYC 7543 CPUs, 512GB DRAM, and NVIDIA Quadro RTX 6000 (24GB) GPUs for experiments of BERT-based models and A100 (80GB) GPUs for Llama2-7B models. The codebase is built on PyTorch 2.1.2, Transformers 4.23.0, PEFT 0.10.0, and Opacus 1.4.1. The source code is attached and should be used with the specified version of Transformers and PEFT packages from HuggingFace and the Opacus library above.

**Impacts of node degree capping.** Table 3 further shows the effects of node degree capping for non-private training. Interestingly, the one with node degree capping usually outperforms the one without node degree capping. We hypothesis that the node degree capping could prevent overfitting to the high-degree nodes and help generalization, as we argued for the adaptive clipping.

Table 2: Dataset statistics and experimental setup for evaluation.

| Dataset | #Entity | #Relation | #Entity (Test) | #Classes | #Relation (Test) | Test Domain |
|---|---|---|---|---|---|---|
| AMAZ-Cloth | 960,613 | 4,626,125 | 476,510 | 9 | 10,000 | AMAZ-Sports |
| AMAZ-Sports | 357,936 | 2,024,691 | 129,669 | 16 | 10,000 | AMAZ-Cloth |
| MAG-USA | 132,558 | 702,482 | 6,653 | 40 | 63,635 | MAG-CHN |
| MAG-CHN | 101,952 | 285,991 | 6,534 | 40 | 34,603 | MAG-USA |

Table 3: Results on zero-shot relation prediction of model fine-tuned with and without degree capping.

| Privacy | Pretrained Models | MAG(CHN→USA) PREC@1 | MRR | MAG(USA→CHN) PREC@1 | MRR | AMAZ(Sports→Cloth) PREC@1 | MRR | AMAZ(Cloth→Sports) PREC@1 | MRR |
|---|---|---|---|---|---|---|---|---|---|
| $\epsilon = \infty$ (no capping) | BERT.base | 28.07 | 39.11 | 41.93 | 53.91 | 36.13 | 47.07 | 29.84 | 39.61 |
| | BERT.large | 26.37 | 37.73 | 40.90 | 53.16 | 36.89 | 47.50 | 29.30 | 39.76 |
| | Llama2-7B | 32.80 | 46.67 | 45.65 | 58.59 | 41.01 | 52.39 | 29.21 | 41.44 |
| $\epsilon = \infty$ (K=5) | BERT.base | 28.92 | 40.05 | 42.34 | 54.58 | 37.47 | 48.22 | 30.13 | 39.98 |
| | BERT.large | 27.67 | 39.23 | 40.40 | 52.79 | 36.25 | 46.90 | 30.75 | 41.36 |
| | Llama2-7B | 39.68 | 53.35 | 46.29 | 59.96 | 38.73 | 50.95 | 32.54 | 45.16 |

# F   Limitations and Future Works

For adaptive clipping, we mainly focused on comparison between clipping by constant (i.e., standard clipping) and clipping by number of node occurrence (which leads to a constant sensitivity). It will be an interesting future direction to further explore the privacy-utility trade-offs of a general adaptive clipping strategy based some functions of node occurrence.

For the privacy amplification bounds of cardinality-dependent coupled sampling, the tightness of the bound was not yet explored, which can be also an interesting future work.

# G   Broader Impacts

This work develops a node-level differentially private DP-SGD framework for relational learning, advancing the theory and practice of privacy-preserving machine learning on graph-structured data. Graph data are ubiquitous in modern machine learning applications, and our work helps protect the sensitive attributes or existence of individuals in a network while still enabling meaningful relational inference.

We do not foresee significant negative societal impacts from this work, since our contribution is theoretical and does not involve the release of high-risk models or data.

