# OpenReview forum: "Differentially Private Relational Learning with Entity-level Privacy Guarantees"
_NeurIPS.cc/2025/Conference — NeurIPS 2025 poster_

### Official Review · Reviewer_LSG2 · 2025-06-29

**Clarity:** 3
**Significance:** 3
**Originality:** 3
**Rating:** 5
**Confidence:** 3

**Summary:**

A fundamental problem that limits the DP-SGD in relational learning is that DP-SGD considers the i.i.d assumption of samples when clipping the gradient and mini-batching. In relational learning, the samples, i.e., the nodes and edges, are dependent. To mitigate this problem, the authors design a new DP-SGD with a specifically designed gradient-clipping strategy and mini-batch construction strategy in this paper.

**Questions:**

1. I think the adaptive clipping scheme relies on per-entity frequency, but the paper does not analyse how the joint distribution of frequencies affects the optimal clip threshold once entities number in the millions (e.g., social graphs). A worst-case entity that appears in most relations could still dominate sensitivity—even with adaptive scaling—forcing very large noise that may erase the reported utility gains. It's might be too complex to present in a single paper, but I'm curious about how the authors think about this problem.

2. Entity-level DP is stronger but costlier than edge-level DP. I'm curious about how much additional noise is paid relative to edge-level privacy.

**Ethical Concerns:**

["NO or VERY MINOR ethics concerns only"]

**Final Justification:**

I think the responses have addressed my questions. And I think the contributions could be vital for related problems in privacy. Therefore, I think this paper should be accepted.

**Limitations:**

No.

**Quality:**

3

**Strengths And Weaknesses:**

Strength:
1. This paper has good presentation quality.

2. The analysis of the problem and the intuition of new methods is comprehensive and makes sense to me.

3. The problem studied by the paper is vital for protecting training data privacy in relational learning.


Weakness:

1. I think the authors should publish their code to increase the reproducibility.

2. Some questions are not answered, please check the questions part.

---

> ### Author Rebuttal · Authors · 2025-07-29
>
> We thank the reviewer LSG2 for their positive feedback. Here is our response.
>
> > **W1**: “The authors should publish their code to increase the reproducibility”
>
> Thank you for the suggestion. We provided code to reproduce all results in the supplementary materials. We will make the code publicly available after the review process
>
> > **Q1**: “I think the adaptive clipping scheme relies on per-entity frequency, but the paper does not analyse how the joint distribution of frequencies affects the optimal clip threshold once entities number in the millions (e.g., social graphs). A worst-case entity that appears in most relations could still dominate sensitivity—even with adaptive scaling—forcing very large noise that may erase the reported utility gains."
>
> It is true that in the worst case, a hub node with a very high degree can appear in all sampled relations in a mini-batch. In our experiments, we mitigate this issue by capping node degrees below a fixed threshold (Line 285-288). We also investigate how this degree capping impacts utility under a fixed privacy budget (Figure 3, middle panel). This may not be the optimal strategy for handling high-degree nodes, and we agree that it is an important direction for future work.
>
> > **Q2**: “Entity-level DP is stronger but costlier than edge-level DP. I'm curious about how much additional noise is paid relative to edge-level privacy.”
>
> That is a good question. We can actually calculate the privacy leakage for edge-level privacy using our framework. Let us denote the standard privacy amplification bound (e.g., Poisson sampling) of sampling rate $\gamma$ by $\epsilon(\gamma)$ (see the bound from Theorem 4 of [1]). We can show that the $\epsilon_{edge}$ parameter of edge-level privacy is equal to $\epsilon(\gamma)$ ($\gamma$ is the sampling rate of positive edge in our context), because only one "row" of positive edge is affected. For a very coarse approximation and comparison here, the $\epsilon_{entity}$ parameter of entity-level privacy (equation (3)) can be roughly approximated by $\epsilon(\Gamma)$ with an “effective” sampling rate $\Gamma:=(K+d\*k_\{neg\})*\gamma$, where $K$ is the maximal node degree, $d$ is the average node degree and $k_{neg}$ is the number of negative sampling per positive. This implies that, compared to edge-level privacy, entity-level privacy incurs an extra factor of $K+d\*k_{neg}$ in its effective sampling rate, which depends on node degree.
>
> [1] Mironov, Ilya, Kunal Talwar, and Li Zhang. "R\'enyi differential privacy of the sampled gaussian mechanism." arXiv preprint arXiv:1908.10530 (2019).

---

> > ### Comment · Reviewer_LSG2 · 2025-08-05
> >
> > Thank you for your detailed response—my concerns have been fully addressed, and I recognise the quality of the work. Nevertheless, because relational learning is a relatively smaller topic compared to LLM, ViT, etc., I believe this limits the paper’s broader significance; accordingly, I will maintain my original score.
> >
> > Please let me know if there are any further questions or knowledge to teach.
> >
> > Finally, I appreciate your commitment to safeguarding user privacy in the current big data era.

---

> > > ### Author Response · Authors · 2025-08-05
> > > **Official Comment by Authors**
> > >
> > > Thank you again for your dedicated time and encouraging comments. We are very grateful to know that our response has fully resolved your concerns.

---

### Official Review · Reviewer_m1UV · 2025-07-02

**Clarity:** 3
**Significance:** 2
**Originality:** 3
**Rating:** 4
**Confidence:** 2

**Summary:**

This paper proposes a DP-SGD framework for relational learning with formal node-level differential privacy guarantees. To address high sensitivity and coupled sampling challenges, the authors introduce adaptive gradient clipping based on node frequency and a new privacy amplification analysis for cardinality-dependent sampling. Experiments on fine-tuning text encoders demonstrate improved privacy–utility trade-offs over standard DP-SGD.

**Questions:**

- This work focuses on node-level DP. Does it offer any insights or potential solutions for achieving edge-level DP in the same context?
- Why is fine-tuning pretrained text encoders chosen as the application? In the task of relational learning, or more specifically, link prediction, GNNs are often directly applied. Is this method suitable for such cases as well?
- In scenarios like online social platform recommendations, where the number of high-degree nodes can be very large, with degrees reaching into the hundreds or thousands, how does degree capping influence performance in such cases?

**Ethical Concerns:**

["NO or VERY MINOR ethics concerns only"]

**Final Justification:**

The authors have clarified the novelty and positioning of their work relative to DP link prediction, and addressed questions regarding applicability and degree capping. As the authors note, the set of available empirical baselines meeting the rigorous entity-level privacy definition is limited. Overall, most concerns have been resolved, and the work makes a meaningful contribution, with the score reflecting its novelty and soundness.

**Limitations:**

Yes.

**Paper Formatting Concerns:**

No major formatting issues.

**Quality:**

3

**Strengths And Weaknesses:**

Strengths
- The paper proposes adaptive gradient clipping and a new amplification analysis tailored for relational learning, offering node-level differential privacy guarantees.
- The proposed method achieves strong privacy–utility trade-offs across various datasets.
- This paper is clearly written, and the tables and charts are easy to understand.

Weaknesses
- The work feels somewhat related to the well-established problem of differentially private link prediction. A discussion highlighting the differences and connections with existing literature on DP link prediction would help clarify the unique contribution of this approach.
- The compared baselines (standard clipping) are limited. Adapting additional recent baselines for comparison would provide a more comprehensive evaluation of the efficacy of the proposed method.

---

> ### Author Rebuttal · Authors · 2025-07-29
>
> We thank the reviewer m1UV for their construcive comments. Here is our response.
>
> > **W1**: “The work feels somewhat related to the well-established problem of differentially private link prediction. A discussion highlighting the differences and connections with existing literature on DP link prediction would help clarify the unique contribution of this approach.”
>
> We thank the reviewer for this constructive suggestion. While link prediction can be considered a special case of relational learning involving binary relations, differentially private link prediction remains an **emerging and relatively underexplored** area. Some recent works have begun to address this topic, such as [1] which focuses on protecting a specific subset of node pairs, and [2] which provides link-level privacy guarantees.
>
> In contrast to these existing approaches, our work offers several key distinctions that highlight its unique contribution:
>
> - Most notably, neither [1] nor [2] addresses entity-level (i.e., node-level) privacy guarantees, which is the core focus of our framework.
> - [2] adopts a binary classification objective function with uniformly sampled positive and negative edges. Our work, however, considers a more general contrastive loss, which involves a more complex process of sampling and grouping negative instances per positive instance.
> - In terms of application, [2] trains graph neural networks for link prediction. Our work, by contrast, focuses on fine-tuning large language models on one relational dataset and then inferring new relations on a different dataset.
>
> We will include a discussion of these related works in the final version of the manuscript.
>
>
> > **W2**: “The compared baselines (standard clipping) are limited. Adapting additional recent baselines for comparison would provide a more comprehensive evaluation of the efficacy of the proposed method.”
>
> We appreciate the reviewer's suggestion. To the best of our knowledge, our approach is the first to provide a formal entity-level privacy guarantee for relational learning. Adapting recent other baselines (which is not for entity-DP relational learning) to meet the rigorous entity-level privacy definition in relational learning is highly nontrivial. Therefore, we focused our comparison against the standard clipping approach adapted to our relational learning setting, which serves as a relevant baseline within this novel problem formulation.
>
> > **Q1**: “This work focuses on node-level DP. Does it offer any insights or potential solutions for achieving edge-level DP in the same context”
>
> While our primary focus is on entity-level (node-level) differential privacy, it can indeed be adapted to provide edge-level DP guarantees within the same context. Entity-level DP considers two neighboring graphs that differ by a single node and all its associated edges. In contrast, edge-level DP considers neighboring graphs that differ by only a single edge. Precisely, the edge-level privacy guarantee of Algorithm 3 can be accordingly provided by Theorem 4.1., with maximal node degree $K$ replaced by 1 (reflecting only one edge can be removed/inserted) and $k_{neg}$ replaced by $0$ (reflecting no nodes is removed/inserted, so no privacy leakage from negative edges).
>
> > **Q2-1**: “Why is fine-tuning pretrained text encoders chosen as the application?”
>
> Broadly, our framework is able to ensure entity-level privacy of relational learning for both fine-tuning pretrained models or training a model from scratch. We chose to focus on fine-tuning pre-trained text encoders because this approach closely aligns with current real-world applications of relational learning. In practice, pre-trained or foundation models are widely utilized for relational data of specific modalities, for example, large language models for textual data in recommendation systems [3], finance [4,6], and healthcare [5], or vision-language models for image data [7].
>
>
>
>
> > **Q2-2**: “In the task of relational learning, or more specifically, link prediction, GNNs are often applied. Is this method suitable for such cases as well?”
>
> While our current method is broadly applicable to relational learning, integrating it directly with GNN architectures introduces additional privacy risks. This is because GNNs involve node feature aggregation via the graph structure. Several existing works have investigated privacy guarantees for GNNs in the context of node classification [8, 9]. Exploring entity-level private relational learning specifically in conjunction with GNN architectures remains a promising direction for future research.
>
>
> > **Q3**: “how does high degree capping influence performance in such cases?”
>
> Degree capping is a tunable hyperparameter that influences the trade-off between utility and privacy. In Figure 3 (middle), we evaluate model utility under varying degree caps while keeping the privacy budget fixed. The results show that utility remains quite robust across different levels of degree capping.
>
> [1] De, Abir, and Soumen Chakrabarti. "Differentially private link prediction with protected connections." Proceedings of the AAAI conference on artificial intelligence. Vol. 35. No. 1. 2021.
>
> [2] Ran, Xun, et al. "Differentially private graph neural networks for link prediction." 2024 IEEE 40th International Conference on Data Engineering (ICDE). IEEE, 2024.
>
> [3] Zhao, Zihuai, et al. "Recommender systems in the era of large language models (llms)." IEEE Transactions on Knowledge and Data Engineering 36.11 (2024): 6889-6907.
>
> [4] Peng, Bo, et al. "eCeLLM: Generalizing Large Language Models for E-commerce from Large-scale, High-quality Instruction Data." Forty-first International Conference on Machine Learning.
>
> [5] Gao, Y., et al. "Leveraging a medical knowledge graph into large language models for diagnosis prediction. arXiv. 2023." arXiv preprint arXiv:2308.14321.
>
> [6] Ouyang, Kun, et al. "Modal-adaptive knowledge-enhanced graph-based financial prediction from monetary policy conference calls with LLM." arXiv preprint arXiv:2403.16055 (2024).
>
> [7] Li, Xin, et al. "Graphadapter: Tuning vision-language models with dual knowledge graph." Advances in Neural Information Processing Systems 36 (2023): 13448-13466.
>
> [8] Daigavane, Ameya, et al. "Node-level differentially private graph neural networks." arXiv preprint arXiv:2111.15521 (2021).
>
> [9] Sajadmanesh, Sina, et al. "{GAP}: Differentially private graph neural networks with aggregation perturbation." 32nd USENIX Security Symposium (USENIX Security 23). 2023.

---

> > ### Comment · Reviewer_m1UV · 2025-08-05
> >
> > Thank you for your rebuttal. Most of my concerns are addressed through your clarifications, and I appreciate your effort in responding to the feedback.
> >
> > After reviewing other reviewers’ perspectives, I will maintain my original score.

---

> > > ### Author Response · Authors · 2025-08-05
> > > **Official Comment by Authors**
> > >
> > > We thank the reviewer's response and the positive feedback on our manuscript. Your thoughtful questions are very helpful in improving our work.

---

### Official Review · Reviewer_xnXK · 2025-07-02

**Clarity:** 4
**Significance:** 3
**Originality:** 3
**Rating:** 5
**Confidence:** 4

**Summary:**

This paper proposes a principled framework for differentially private relational learning, focusing on edge existence as the sensitive attribute. It introduces adaptive gradient clipping and a novel sampling analysis to handle high sensitivity and interdependent data and uses these to introduce an new DP-SGD framework suited to relational learning.

**Questions:**

- Would the performance improvement be as significant in the high privacy regime?
- Can you provide an upper bound for ε(α) in the theorem to help with its interpretation?

**Ethical Concerns:**

["NO or VERY MINOR ethics concerns only"]

**Limitations:**

Yes

**Quality:**

3

**Strengths And Weaknesses:**

Strengths:
- The paper addresses a timely and previously unexplored problem, contributing novel insights to the field.

- It presents a comprehensive critique of existing approaches, clearly articulating their limitations. The proposed framework demonstrates significantly improved privacy guarantees compared to naive baselines.

Weaknesses:
-The experimental evaluation lacks results for high-privacy regimes (e.g., epsilon = 1), which limits the assessment of the framework's performance under stricter privacy constraints.

---

> ### Author Rebuttal · Authors · 2025-07-29
>
> We thank the reviewer xnXK for their positive feedback.
>
> > **W1 & Q1**: “The experimental evaluation lacks results for high-privacy regimes”
>
> We thank the reviewer for the comment. Our experimental evaluation, specifically the rightmost panel in Figure 3, already studies the utility-privacy trade-offs across a range of privacy budgets, from $\epsilon>10^5$ down to $\epsilon=2$. This range of $\epsilon$ is consistent with previous works in node-DP graph learning, such as [1]. Generally, achieving very strong node-level (entity-level) differential privacy, such as setting $\epsilon=1$, can lead to significant degradation in model utility, as observed in other node-DP literature (e.g., [2], Figure 6). We believe the chosen range adequately demonstrates the trade-offs of our proposed method.
>
>
>
> > **Q2**: “An upper bound for $\epsilon(\alpha)$ to help with its interpretation”
>
> The amplification bound in Equation (3) admits a natural interpretation: the $\Psi$ term corresponds to the exponential of the Rényi divergence between a Gaussian mixture and a zero-mean Gaussian. This mirrors the form of the standard Rényi differential privacy amplification bound [3], but with a modified effective sampling rate that depends on node degree. The upper bounds presented in [3] are applicable to our setting and can be used to estimate the divergence, albeit with some loss in tightness.
>
>
> [1] Daigavane, Ameya, et al. "Node-level differentially private graph neural networks." arXiv preprint arXiv:2111.15521 (2021).
>
> [2] Sajadmanesh, Sina, et al. "{GAP}: Differentially private graph neural networks with aggregation perturbation." 32nd USENIX Security Symposium (USENIX Security 23). 2023.
>
> [3] Mironov, Ilya, Kunal Talwar, and Li Zhang. "R\'enyi differential privacy of the sampled gaussian mechanism." arXiv preprint arXiv:1908.10530 (2019).

---

### Official Review · Reviewer_nM78 · 2025-07-22

**Clarity:** 2
**Significance:** 2
**Originality:** 2
**Rating:** 2
**Confidence:** 4

**Summary:**

This paper introduces a novel framework for relational learning with entity-level differential privacy (DP) guarantees, addressing challenges unique to graph-structured data.  Relational learning, which leverages relationships between entities, is increasingly applied in sensitive domains like healthcare and finance, where privacy concerns are paramount.  Standard DP mechanisms, such as DP-SGD, struggle to handle relational data due to high sensitivity caused by entities participating in multiple relations and coupled sampling processes that complicate privacy amplification analyses.  To overcome these challenges, the authors propose a tailored DP-SGD variant that incorporates adaptive gradient clipping based on entity occurrence frequency and a privacy amplification analysis for a tractable subclass of coupled sampling. These innovations ensure rigorous privacy guarantees while maintaining strong utility in relational learning tasks.

The framework is validated through experiments on fine-tuning pre-trained text encoders using text-attributed relational datasets.  Results demonstrate that the proposed method achieves better privacy-utility trade-offs compared to standard DP-SGD approaches.  The paper also provides detailed sensitivity analyses, theoretical proofs, and privacy amplification bounds, ensuring the robustness of the proposed method.  Additionally, the authors explore the impact of key parameters, such as node degree capping and the number of negative samples, on model performance and privacy guarantees.  By advancing the theory and practice of privacy-preserving machine learning for graph data, this work contributes to safeguarding sensitive information while enabling meaningful relational inference in real-world applications.

**Questions:**

Please focus on addressing the weaknesses raised above.

**Ethical Concerns:**

["NO or VERY MINOR ethics concerns only"]

**Limitations:**

Yes.

**Quality:**

2

**Strengths And Weaknesses:**

## Strengths

- Extensive theoretical analysis and experimental results are conducted on the proposed method, providing a thorough study of the method.
- The proposed method provides an optimization process for relational learning under strong privacy protection of node-level DP.
- The proposed mechanism can be easily adapted from an existing mechanism (DP-SGD).

## Weaknesses

- The analysis of local sensitivity is conducted on a scenario where any node $u^*$ sampled in an edge tuple $B$ must only exist in one edge tuple $T_i$ (based on lines 157-160). However, the proposed mechanism never discusses how to ensure this happens in Algorithms 1, 2, or 3. Furthermore, in this scenario, the sampling of the batch $B$ does not follow Poisson sampling anymore because it has to ensure any node $u$ only appears in one edge tuple $T_i$, leading to the uncertainty of the privacy guarantee of Alg. 3.
- Although the proposed method leverages an adaptive clipping method, it is not very efficient. Specifically, in the clipping process of Alg. 2, after adding the $max-freq(T_i)$ to the clipping threshold, it seems like the new threshold is always smaller than $C$, incurring more information loss from the gradient. Furthermore, the noise added to the gradient is still $N(0, \sigma^2C^2I)$, which is not scaled to the local sensitivity. Although the DP protection stays the same as the original DP-SGD, the information loss is higher due to the lower gradient clipping threshold.
- The privacy application analysis in Theorem 4.1 ignores the impact of multiple training rounds $T$. This is a critical point because the training process is conducted with multiple training rounds, which incur more privacy amplification as discussed in DP-SGD.
- The experiments do not consider other state-of-the-art baselines for both node-level DP (e.g., Sajadmanesh et al. [1]) and entity-level DP (e.g., Lai et al. [2]).
- The presentation of the paper is hard to follow. The mathematical proofs of the theorems and propositions are very confusing, with lots of notation that was never introduced in the main body.

[1] Sajadmanesh, Sina, et al. "{GAP}: Differentially private graph neural networks with aggregation perturbation." 32nd USENIX Security Symposium (USENIX Security 23). 2023.

[2] Lai, Phung, et al. "User-entity differential privacy in learning natural language models." 2022 IEEE International Conference on Big Data (Big Data). IEEE, 2022.

---

> ### Author Rebuttal · Authors · 2025-07-29
>
> We thank Reviewer nM78 for their comments. Here is our response.
>
> > **W1-1**: “The analysis of local sensitivity is conducted on a scenario where any node $u^\{\*\}$ must only exist in one edge tuple $T_i$ (based on lines 157-160). However, the proposed mechanism never discusses how to ensure this happens in Algorithms 1, 2, or 3”
>
> We thank the reviewer for their valuable feedback. We apologize for the lack of clarity in our manuscript that led to confusion regarding our sensitivity analysis. We would like to clarify that our framework **does not assume** a node exists in only one edge tuple; on the contrary, it is designed to handle the general case where a node can appear in an **arbitrary number** of edge tuples.
>
> The passage on lines 157-160, which describes when a "node $u^\{\*\}$ appears only in a negative edge," serves to formally define a neighboring mini-batch for the purpose of our analysis. It illustrates the impact of removing $u^\{\*\}$ on a specific subset of edge tuples (denoted by $B_\{-\}(u)$) whose negative edges contain $u^*$. The fact that an entity can be associated with an arbitrary number of edge tuples is precisely what leads to the high sensitivity that we aim to resolve. Our analysis captures this explicitly in Equation (2), which sums the gradient contributions over all tuples containing the node $u^\{\*\}$.
>
>
> To manage this high sensitivity, our work proposes two key strategies:
> - Adaptive Clipping (Algorithm 1): We introduce FREQ-CLIP, which calculates the occurrence frequency of each node within the mini-batch and uses this frequency to modulate the gradient clipping threshold. This directly counteracts the large influence of nodes that appear in many tuples.
> - Constraining Negative Occurrences (Algorithm 2): We propose to use a specific negative sampling strategy, NEG-SAMPLE-WOR, which samples nodes without replacement. This design is crucial as it guarantees that any given node can appear in the negative edges of at most one tuple within the batch (i.e., $|B_\{-\}​(u)|\le 1$ ).
>
> As demonstrated in Proposition 4.1, the combined use of these two strategies successfully bounds the global sensitivity to a constant, which is a key requirement for DP-SGD.
>
>
> > **W1-2**: "Furthermore, in this scenario, the sampling of the batch $B$ does not follow Poisson sampling anymore because it has to ensure any node only appears in one edge tuple, leading to the uncertainty of the privacy guarantee of Alg. 3."
>
> It is indeed correct that the batch $B$ does not solely follow Poisson sampling. To control batch sensitivity and ensure a rigorous privacy guarantee, our approach involves a specific negative sampling strategy, such as sampling without replacement, which ensures that each node appears in at most one negative edge within a mini-batch. The mini-batch $B$ is constructed through a two-step sampling process (see lines 143–147), which we formally define as 'Coupled Sampling' in Definition 4.1. A key contribution of our work is providing a **rigorous privacy amplification guarantee** for this coupled sampling scheme (Theorem 4.1), which directly quantifies the privacy leakage of Algorithm 3.
>
>
> > **W2**: “it seems like the new threshold is always smaller than $C$ , incurring more information loss from the gradient. Furthermore, the noise added to the gradient is still $N(0, \sigma^2C^2I)$ , which is not scaled to the local sensitivity”
>
> We appreciate the reviewer's insightful observation that the adaptive clipping thresholds can be smaller than $C$. The adaptive clipping thresholds indeed vary for different edge tuples within a data batch, and they are designed to be potentially smaller than $C$ to ensure the **overall batch sensitivity** remains bounded by $C$. The noise added to the batch gradient is scheduled according to this overall batch sensitivity $C$, ensuring that the clipping threshold appropriately matches the injected noise.
>
> Regarding the concern about potential information loss: while the gradient clipping threshold is lower for some high-degree nodes, these nodes are also sampled more often during training. This repeated exposure could help compensate for any potential information loss from individual gradient terms. Furthermore, as discussed in lines 195–202, down-weighting the gradients of high-degree nodes can actually help mitigate overfitting and improve generalization to low-degree nodes. Our empirical results in Table 1 indeed demonstrate that adaptive clipping achieves a better utility-privacy trade-off compared to standard gradient clipping, which clips all gradient terms using a constant threshold.
>
>
>
> > **W3**: “The privacy amplification analysis in Theorem 4.1 ignores the impact of multiple training rounds”
>
> Privacy amplification Theorem 4.1 calculates a **per-round** privacy bound, and we calculate the overall privacy bound for multiple training rounds by RDP **composition theorem** ([3], Proposition 1). This is common practice for privacy amplification analysis, e.g., [4]. Thank you for making this point, and we will highlight this as a per-round bound in the revised manuscript.
>
>
> > **W4**: “The experiments do not consider other state-of-the-art baselines for node-DP [1,2]”
>
> We appreciate the reviewer's suggestion to consider additional state-of-the-art baselines for node-DP, such as those in [1] and [2]. However, it's important to note that these methods are **not directly applicable** to providing entity-level differential privacy guarantees within the context of relational learning. This is due to the unique technical challenges presented by the relational learning training objective (Equation (1)). Specifically, relational learning introduces complex sensitivity issues (Equation (2)) and a distinct privacy analysis problem for coupled sampling (Definition 4.1).
>
> In contrast, [1] primarily focuses on node classification tasks with Graph Neural Networks (GNNs), where privacy mechanisms prevent leakage of node features during neighborhood aggregation. [2] addresses user privacy in language model training on textual data. Neither of these approaches is designed to handle the core privacy risk in relational learning, where relations themselves are part of the loss function computation and mini-batches are constructed through complex, coupled sampling of relations. We provided a more detailed discussion on why existing DP-GNN methods are not applicable in our Related Work section (lines 68–81).
>
>
>
> [1] Sajadmanesh, Sina, et al. "{GAP}: Differentially private graph neural networks with aggregation perturbation." 32nd USENIX Security Symposium (USENIX Security 23). 2023.
>
> [2] Lai, Phung, et al. "User-entity differential privacy in learning natural language models." 2022 IEEE International Conference on Big Data (Big Data). IEEE, 2022.
>
> [3] Mironov, Ilya. "Rényi differential privacy." 2017 IEEE 30th computer security foundations symposium (CSF). IEEE, 2017.
>
> [4] Zhu, Yuqing, and Yu-Xiang Wang. "Poission subsampled rényi differential privacy." International Conference on Machine Learning. PMLR, 2019.

---

> > ### Comment · Reviewer_nM78 · 2025-08-05
> >
> > Thanks a lot for the rebuttal. It only partially addressed my questions. I maintain my score.

---

> > > ### Author Response · Authors · 2025-08-05
> > > **Would you mind letting us know your unresolved questions?**
> > >
> > > Thank you for your response. We believe we tried to provide a detailed response to each of your points and clarified the key aspects of our work. To help us better understand and address your remaining concerns, could you please specify which of your questions you feel were not fully resolved? Thank you again for your time and feedback.

---

### Decision · Program_Chairs · 2025-09-17

**Decision:**

Accept (poster)

**Comment:**

The manuscript aims to adapt differentially private stochastic gradient descent (DP-SGD) to relational learning tasks where each node corresponds to a entity/user and preserve entity-level/node-level DP in low privacy settings. The theoretical and experimental results are appreciated [nM78, m1UV, LSG2] except rather high privacy parameters [xnXK] ](eps >= 2 implies that the model would be over 7x more/less likely in case of the presence/absence of a data point). While more baselines were desired [nM78, m1UV], the studied problem was considered very relevant [nM78, xnXK].